# Vinculin is required for interkinetic nuclear migration (INM) and cell cycle progression

Andrea Ochoa[1]* , Antonio Herrera[1]* , Anghara Menendez[1] , María Estefanell[1] , Carlota Ramos[1] , and Sebastian Pons[1]

**Vinculin is an actin-binding protein (ABP) that strengthens the connection between the actin cytoskeleton and adhesion complexes. It binds to β-catenin/N-cadherin complexes in apical adherens junctions (AJs), which maintain cell-to-cell adhesions, and to talin/integrins in the focal adhesions (FAs) that attach cells to the basal membrane. Here, we demonstrate that β-catenin targets vinculin to the apical AJs and the centrosome in the embryonic neural tube (NT). Suppression of vinculin slows down the basal-to-apical part of interkinetic nuclear migration (BA$^{INM}$), arrests neural stem cells (NSCs) in the G2 phase of the cell cycle, and ultimately dismantles the apical actin cytoskeleton. In the NSCs, mitosis initiates when an internalized centrosome gathers with the nucleus during BA$^{INM}$. Notably, our results show that the first centrosome to be internalized is the daughter centrosome, where β-catenin and vinculin accumulate, and that vinculin suppression prevents centrosome internalization. Thus, we propose that vinculin links AJs, the centrosome, and the actin cytoskeleton where actomyosin contraction forces are required.**

## Introduction

In the embryonic neural tube (NT), neural stem cells (NSCs) form a pseudostratified, single-cell layered epithelium called the neuroepithelium that extends from the ventricle to the basal lamina and displays a marked apicobasal polarity. During the early stages of development, NSCs nuclei are densely packed and occupy different positions along the apicobasal axis while proliferating in a self-expanding mode due to a phenomenon known as interkinetic nuclear migration (INM; Sauer, 1936). Later on, coordinated with the onset of neurogenesis, their mode of division changes to generate the first committed neurons, whose nuclei migrate to the basal side of the NT before neuron delamination (Götz and Huttner, 2005; Saade et al., 2013). The nuclei position must be thoroughly regulated during the cell cycle and neurogenesis for the correct development of the central nervous system (CNS). INM refers to the fact that although all cells in pseudostratified epithelia extend from the luminal surface to the basal lamina, their nuclei move to different apicobasal positions throughout the cell cycle; nuclei in G1/S phases move from apical to basal (AB$^{INM}$) and from basal to apical during G2 (BA$^{INM}$). Under normal conditions, mitosis always takes place at the apical surface (Taverna and Huttner, 2010). Pseudostratified epithelia can be divided into short, intermediate, and long depending on how many layers of nuclei make up the epithelium, and, although the INM operating in all of them presents many similarities, there are also important differences depending on the epithelia thickness (Norden, 2017). The issue

that has generated the most controversy when establishing a common model that would explain the functioning of the INM in the different epithelia has undoubtedly been the origin of the forces that move the nuclei in each part of the INM (Bertipaglia et al., 2018; Strzyz et al., 2016). Although not all published work may entirely agree, it is widely accepted that the main force driving INM in shorter epithelia (zebrafish retina and hindbrain) is generated by actomyosin contraction acting during BA$^{INM}$ (Leung et al., 2011; Norden et al., 2009; Yanakieva et al., 2019), with the AB$^{INM}$ a passive stochastic movement resulting from the pressure generated by the cells doing the BA$^{INM}$ (Kosodo et al., 2011; Leung et al., 2011; Norden et al., 2009). In contrast, in longer epithelia like the embryonary rodent cortex, INM seems to depend on tubulin-bound dynein also acting during BA$^{INM}$ (Hu et al., 2013; Schenk et al., 2009; Tsai et al., 2010). Although the involvement of actomyosin cytoskeleton in INM has been reported in many epithelia, the mechanism by which the actomyosin forces may push and/or pull the nuclei is still poorly understood. In zebrafish retina, actomyosin that concentrates basal to nuclei generates pushing forces during BA$^{INM}$; in contrast, in straight neuroepithelia such as zebrafish hindbrain and chicken NT, this basal actomyosin concentration has not been observed (Yanakieva et al., 2019).

The actin cytoskeleton is an essential component of eukaryotic cells that plays an important role in many basic biological functions. In the NT, the actin cytoskeleton allows the elongated

.............................................................................................................................................................................................

[1]Instituto de Biología Molecular de Barcelona (CSIC), Barcelona, Spain.

*A. Ochoa and A. Herrera contributed equally to this paper.  Correspondence to Antonio Herrera: antonio.herrera@epfl.ch;  Sebastian Pons: spfbmc@ibmb.csic.es.

shape of neuroepithelial cells by anchoring the actin filaments (F-actin) to the apical adherens junctions (AJs) and basal focal adhesions (FAs) through N-cadherin/β-catenin and integrins, respectively. In both cases, the anchorage is mediated by a variety of proteins known as actin-binding proteins (ABPs) that strengthen the binding of the actin cytoskeleton to adhesion complexes, especially during actomyosin contraction. In a recent work, we combined protein affinity purification with mass spectrometry to identify the proteins that bind to N-cadherin through its β-catenin binding domain in the chicken NT (Herrera et al., 2021). Here, we show that among the six ABPs identified in that study, up to four (α-N-catenin, cortactin, DBNL, and vinculin) are to some degree necessary to maintain the actomyosin cytoskeleton. However, we found that in this context, vinculin develops functions that go far beyond its function as a passive reinforcement of the actomyosin structure. In this report, we shed light on the critical role of vinculin in NSCs of developing neuroepithelial tissue. Our findings demonstrate that vinculin accumulates primarily at the apical AJs and centrosomes, where it interacts with β-catenin. We observed that vinculin knockdown impairs BA$^{INM}$ and delays G2 to M progression. Additionally, suppression of vinculin hinders the internalization of centrosomes during G2, thereby delaying the gathering of the centrosome with the nucleus, a key event that triggers mitosis in pseudostratified epithelia. Our data further show that vinculin-mediated regulation of BA$^{INM}$ and G2-phase progression depends on its interaction with β-catenin at the apical AJs and centrosomes. We propose that vinculin serves as a critical link between the AJs, the centrosome, and the actin cytoskeleton, providing precise actomyosin-driven forces essential for proper NSC proliferation and differentiation.

## Results

### Vinculin is required to maintain the actin cytoskeleton and the structure of the neuroepithelium

In a previous work, we demonstrated that β-catenin plays a crucial role in cell polarization during neuroepithelium development (Herrera et al., 2014). More recently, we combined protein affinity purification with mass spectrometry to identify the proteins that associate with N-cadherin through its β-catenin binding domain, either directly or through β-catenin (Herrera et al., 2021). Among others, we identified up to six ABPs (Fig. 1 A). The actin cytoskeleton plays a fundamental role in the stability of the apical complex (AC), cell polarity, and global neuroepithelium structure (Arai and Taverna, 2017); therefore, we studied the role of this group of ABPs in the neuroepithelium development. We used in ovo electroporation of shRNAs-generating constructs to individually suppress the different ABPs (Fig. 1 B). We produced two constructs for each gene, tested them in chicken embryonic fibroblast (CEF) cultures (Fig. S1, A–F; Herrera et al., 2021), and the most efficient of each pair was transfected in ovo for 48 h into the NTs of Hamburger and Hamilton state 12 (HH12) chicken embryos (Hamburger and Hamilton, 1992). We stained transversal sections of the embryos with phalloidin to reveal the actin cytoskeleton, whereas GFP expression indicated cell transfection.

For each shRNA, we quantified the percentage (%) of slices showing major alterations in the actin cytoskeleton (gaps in the apical actin domain representing >10% of the transfected area): α-actinin4 and α-E-catenin 0%, α-N-catenin 65%, cortactin 62%, DBNL 55%, and vinculin 98% (Fig. 1, C–J). Actin apicobasal distribution plots confirmed the previous results (Fig. 1, D′–J′). Although suppression of α-N-catenin, cortactin, or DBNL all caused aberrant apicobasal actin distribution, it was the suppression of vinculin that caused the greatest alterations in both the actin cytoskeleton and the structure of the neuroepithelium. Vinculin suppression affected the actin cytoskeleton all along the apicobasal length of the neuroepithelium; however, the effect was by far more evident in the apical pole, where the characteristic actin accumulation forming mainly the apical actin belt was entirely erased (Fig. 1 J′). Consistently, the overexpression of wild-type mouse vinculin (mVCL) led to the accumulation of actin at the apical pole and the protrusion of actin rings into the ventricle. On the other hand, the expression of a vinculin mutant that does not bind actin (mVCL-ΔEx20) resulted in a loss of tension at the apical pole, leading to the invasion of cells into the ventricle (Fig. S2).

### Apical vinculin accumulates at AJs and the daughter centriole of NSCs

Vinculin is an essential component for cell adhesion; it binds to α/β-catenin to form complexes with cadherins in AJs and, through talin, binds to integrins in FAs (Bays and DeMali, 2017). In both situations, vinculin exerts a mechanical force, which helps stabilize adhesion. While its interaction with integrins at FAs has been described in a wide range of cell types and tissues, few studies report its binding to AJs (Bays and DeMali, 2017; Carisey and Ballestrem, 2011). To our knowledge, there are no reports describing vinculin distribution in the NT. However, it was observed in the apical AJs in the *Xenopus* neural plate (Roffers-Agarwal et al., 2008) and was assumed to be present at the FAs that anchor the neuroepithelium to the basal membrane during NT development (Long et al., 2016). By in situ hybridization, we observed that vinculin mRNA expression was abundant in HH18 chicken NT (Fig. 2 A). As none of the antibodies against vinculin that we tested turned out to be appropriate to study endogenous vinculin distribution in chicken NT slices, to infer it, we transfected HH12 chicken embryos with low concentration vinculin$^{T12}$·mCherry construct, a well-characterized vinculin mutant (Cohen et al., 2005; Huang et al., 2017); although we were conscious that the localization results obtained with this vinculin mutant had to be taken with caution due to its reduced inhibitory self-interaction, we believed it could be very useful to predict the main vinculin location sites in chick embryonic NT. Notably, 24 h after electroporation (hpe), Vinculin$^{T12}$·mCherry signal was especially intense in the apical AJs colocalizing with N-cadherin (Fig. 2, B and C). Besides, abundant endogenous vinculin was copurified associated with β-catenin·ST (Fig. 2 D). β-catenin is required for centrosome splitting during mitosis; its deficiency causes monopolar spindles (Kaplan et al., 2004). During interphase, β-catenin is positioned between the two centrosomes, forming a complex with Nek2 and its substrates C-Nap1 and rootletin. Rootletin forms a

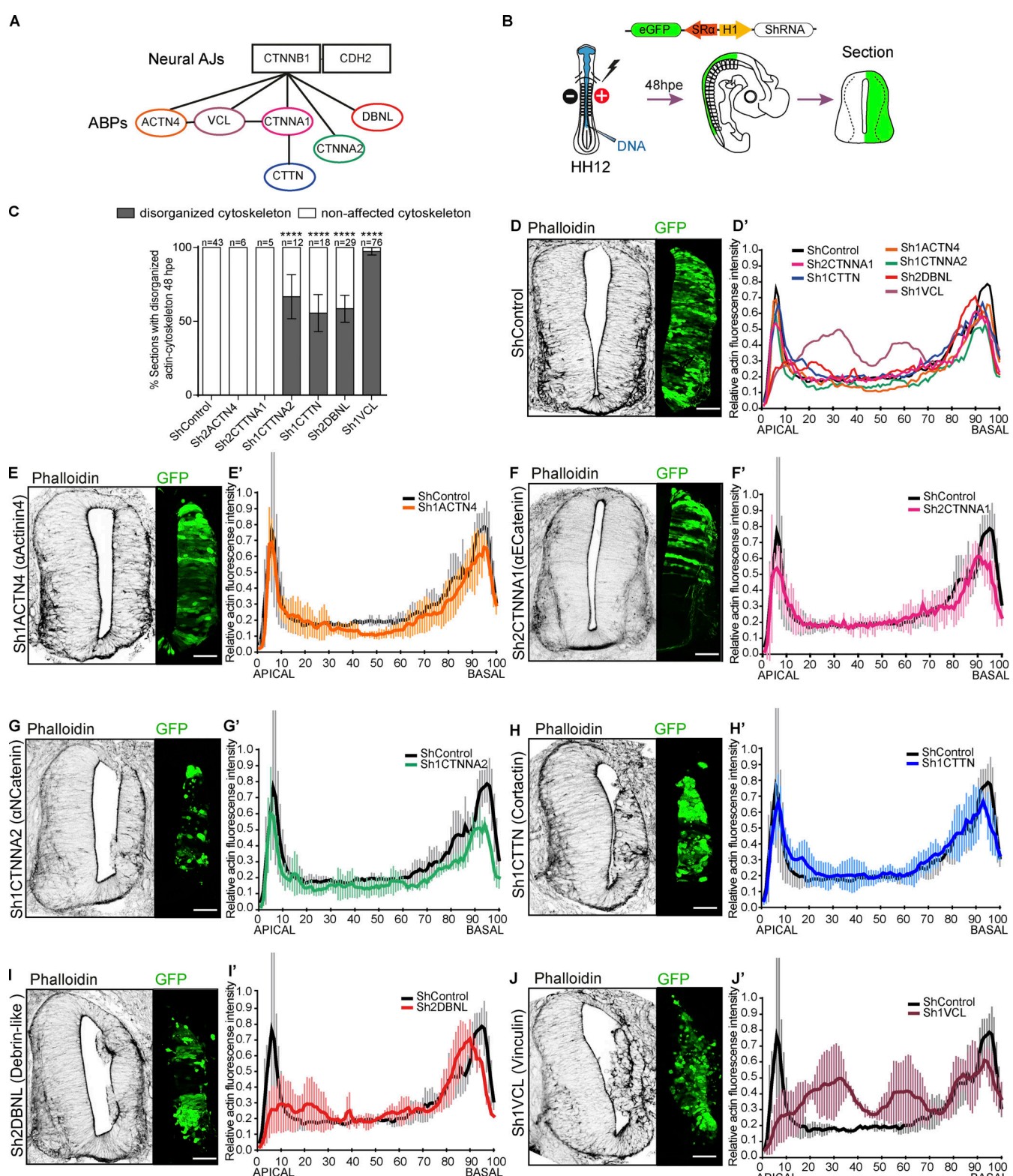

Figure 1. **Vinculin is required to maintain the actin cytoskeleton and the structure of the neuroepithelium. (A)** Summary of actin-binding proteins (ABPs) associated with neural adherens junctions (AJs, CTNNB1[β-catenin]/CDH2 [N-cadherin]) detected by LS-MS/MS. ACTN4 (actinin-4), VCL (vinculin), CTNNA1 (α-E-catenin), CTNNA2 (α-N-catenin), CTTN (cortactin), and DBNL (Dbnl). **(B)** The scheme of the procedure followed in the experiments is shown in Fig. 1, C–J. The NTs of stage HH12 chicken embryos (48 h of incubation) were transfected for 48 h (to stage HH23) with vectors expressing the shRNAs designed to silence the different ABPs and GFP as a reporter of transfection; hpe, hours after electroporation. **(C)** Quantification of the percentage of sections with disorganized actin cytoskeleton after electroporation with the different shRNA vectors. Mean ± SEM. One-way ANOVA plus Tukey's multi-comparisons test, $n$ = 6–76 sections from at least two independent experiments using three or four embryos per condition. **(D–J)** Phalloidin staining (actin, gray scale) of transverse sections of stage HH23 chicken NTs electroporated at stage HH12 for 48 h with the different shRNA vectors. GFP indicates transfection (green).

Scale bar = 50 µm. **(D′–J′)** Plot profiles showing actin distribution along the apicobasal axis (relative actin fluorescence intensity) after electroporation of the different shRNAs. The profiles are shown all together in D′ plot, or individually with the control profile in the rest of the plots. Mean ± SD, $n$ = 7 sections from two independent experiments using at least three embryos per condition. **** = P < 0.0001.

bridge between the two centrosomes by binding to C-Nap1, which is located in the proximal part of the centrioles. At the onset of mitosis, the activity of Nek2 increases, leading to the disassembly of the rootletin bridge and causing the binding of β-catenin to rootletin-independent sites on the centrosomes (Bahmanyar et al., 2008). In the NT, β-catenin is an apical protein that has been found to localize in adherens junctions and centrosomes (Chilov et al., 2011; Lee and Norden, 2013; Taverna and Huttner, 2010). However, there is currently no reliable evidence indicating the presence of vinculin in centrosomes. To obtain a more accurate representation of the relationship between vinculin and β-catenin at the apical side, we transfected HH12 NTs with β-catenin-FLAG and wild-type vinculin·mCherry and employed open book preparations to capture "en face" images (Fig. 2 E). By projecting the seven Z-planes containing the apical domain, we observed that apart from the expected colocalization at the AJs already predicted by the Vinculin[T12]·mCherry construct, vinculin colocalized with β-catenin at the centrosome, which was visualized by the transfection of Cep152·GFP (Fig. 2 F). We then quantified the fluorescence intensity of each protein in a rectangular area containing the two centrioles. The resulting profiles showed that the three proteins had mostly overlapping patterns. However, while the Cep-152 profile displayed two peaks, the profiles of vinculin and β-catenin had a single coincident peak that did not entirely overlap with either of the two Cep-152 peaks (Fig. 2 F′). To gain a more detailed understanding of the Z-axis distribution of the proteins, we also measured the mean fluorescence intensity of each protein across 37 Z-planes. Notably, while the three proteins coincided in the most apical planes, β-catenin and, in particular, vinculin remained more elevated than Cep152 in deeper Z-planes (Fig. 2 F″). To establish the precise location of β-catenin/vinculin complex with respect to the mother and daughter centrioles, we electroporated NTs with β-catenin-FLAG, Cep152·GFP, and Arl13b-RFP. Interestingly, we observed that β-catenin had a ring-shaped distribution that was located distally with the Cep-152 staining of the daughter centriole (Fig. 2 G and Video 1). All these results demonstrate that the apical β-catenin/vinculin complex is associated with AJs and the daughter centriole in NSCs.

**Vinculin knockdown induces accumulation of NSCs nuclei at the basal side of the neuroepithelium before affecting the apical actin cytoskeleton**

In view of the profound effects that the reduction of vinculin had on the structure of neuroepithelium at 48 hpe, we transfected chicken embryos at stage HH12 for 24, 36, and 48 h with shVCL and studied their actin distribution (Fig. 3 A). Both the actin cytoskeleton and the neuroepithelial structure were intact at 24 hpe; however, frequent tissue malformations were already observed at 36 hpe (Fig. 3 B). This result indicated that the pool of vinculin present after electroporation was consumed until its

concentration was insufficient to maintain the actin cytoskeleton. However, to rule out that the effect was not due to different vinculin requirements along development, we transfected embryos at stages HH18 and HH23 for 24 h with shVCL (Fig. 3 C). Supporting the consumption hypothesis, in this case, neither the actin cytoskeleton nor the neuroepithelial structure was affected in any of the transfected embryos (Fig. 3 D). As mentioned above, no defects in the apical actin cytoskeleton were noticeable 24 h after vinculin suppression; however, an abnormal accumulation of cell nuclei seemed to take place in the basal side of the epithelium. In neuroepithelia, during the G1 phase, the nuclei migrate from apical to more basal positions where they perform the S phase; later, during the G2 phase, they return to the apical pole where mitosis finally takes place (Lee and Norden, 2013; Taverna and Huttner, 2010; Fig. 3 E). Notably, we confirmed that the mean nuclei position was indeed shifted toward the basal side 24 h after vinculin knockdown with shVCL (median of 69.96% for shVCL and 52.20% for control, where 0% and 100% represent the apical and the basal membranes, respectively; Fig. 3, F and G). As development proceeds, neurogenic divisions in the ventricular zone produce neurons that migrate laterally, creating the mantle zone in a process called delamination (Fig. 3 E). Therefore, we contemplated the possibility that the basal nuclei accumulation could be due to premature differentiation/delamination of NSCs; however, the percentage between progenitors (Sox2[+]) and neurons (HuC/D[+]) was not affected by vinculin suppression (Fig. 3, H–J), ruling out that possibility.

**Vinculin is required for BA[INM]**

The results shown so far suggested that the lack of vinculin could be affecting INM. In support, studies realized in pseudostratified epithelia from different origins conferred to actin an important role in INM. Despite this, there is still controversy about the involvement of actin in each of the two legs of the INM; AB[INM] and BA[INM] (Lee and Norden, 2013; Taverna and Huttner, 2010). To study the contribution of actin cytoskeleton and vinculin in chicken NT INM, we took time-lapse confocal images of NT ex-vivo cultures transfected for 8–10 h with a control vector or shVCL or treated with blebbistatin (a myosin II inhibitor). Membrane-GFP and H2B-RFP were used to track the contour and the nucleus of transfected cells, respectively. In each case, 18 cells moving apical-to-basal and 18 moving basal-to-apical were tracked (X, Y positions versus time) for as long as possible; representative condensed sequences and drawings illustrating the main movement features of each treatment and direction are shown. The corresponding full videos can be found as supplementary information (Fig. 4, A–F; and Videos 2, 3, and 4 for AB[INM]; and Fig. 5, A–F; and Videos 5, 6, and 7 for BA[INM]). In addition, we plotted the X components of trajectories (corresponding to apicobasal positions, Y axis), against time (X axis), to generate trajectory plots for all the mentioned conditions

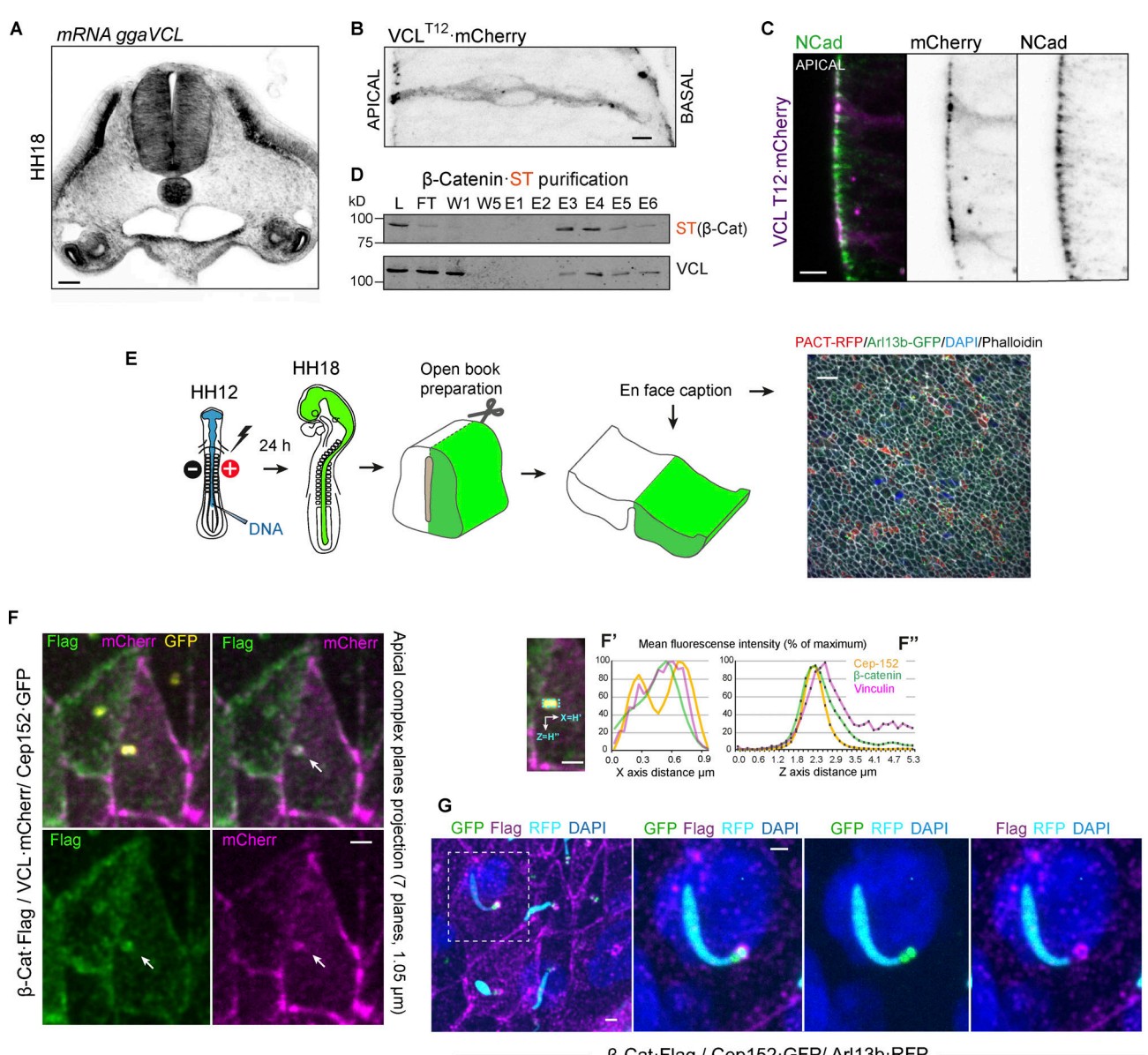

**Figure 2. Vinculin accumulates at apical pole of NSCs with the AJs. (A)** Image of a chicken NT transverse section at stage HH18 showing the expression of vinculin mRNA by in situ hybridization. Scale bar = 50 μm. **(B)** High magnification image spanning the entire apicobasal length showing a NT at stage HH18 24 hpe with VCL-T12·mCherry (gray scale). Scale bar = 10 μm. **(C)** Apical region of an NT slice as in B, stained with anti-N-cadherin (green) and VCL-T12·mCherry (magenta), channels are shown separately at the right-hand panels (gray scale). Scale bar = 10 μm. **(D)** ST-tagged β-catenin was electroporated for 24 h into stage HH12 chicken NTs. At HH18, β-catenin·ST and the endogenous associated proteins were purified on Streptactin columns. The presence of ST (β-catenin) and vinculin in the elution fractions was assessed by Western blot. L, lysate; FT, flow through; W, wash; E, elution. **(E)** Drawing of the procedure followed to obtain "en face" pictures of the apical pole of neural precursors. A low magnification image of a test electroporation is shown. Centrioles, primary cilia, actin, and nucleus are labeled with PACT, Arl13b, phalloidin, and DAPI, respectively. Scale bar = 10 μm. **(F)** "En face" pictures of chicken NTs transfected at stage HH12 for 24 h with Cep152-GFP (yellow), VCL·mCherry (magenta), and β-Catenin·flag (green). Different combinations of channels are shown for clarity. Arrows indicate the centrioles position. The images are Z projections of seven confocal planes spaced 0.15 μm. Scale bar = 1 μm. **(F')** X-axis projections of the fluorescence intensity of a rectangular area (cyan dotted rectangle) that contained the centrioles. **(F")** The mean fluorescence intensity of the mentioned area was calculated in 37 consecutive confocal planes spaced 0.15 μm. Scale bar = 1 μm. **(G)** "En face" images of chicken NTs transfected at stage HH12 for 24 h with Cep152-GFP (green), β-Catenin·Flag (magenta), and Arl13b-RFP (cyan). Nuclei were stained with DAPI. The encircled area in the left-handed panel is enlarged in the right-handed panels, where different channel combinations are shown for clarity. Scale bar = 1 μm. Source data are available for this figure: SourceData F2.

(Fig. 4, G–I for AB$^{INM}$ and Fig. 5, F–I for BA$^{INM}$). In addition, we calculated the mean velocity (mV, Fig. 6, A–A"), the directional velocity (dV, Fig. 6, B–B"), and the directionality ratio (dR, Fig. 6, C–C") of each cell. In control conditions, as in other models (Leung et al., 2011; Norden et al., 2009), AB$^{INM}$ was slower and less directed than BA$^{INM}$ (Fig. 6, A'–C" and Table S1). Next, we studied AB$^{INM}$ and BA$^{INM}$ kinetics after myosin II inhibition with blebbistatin at 12.5 μM, and at this concentration, blebbistatin reduces INM but does not affect actin cytoskeleton stability (Schenk et al., 2009). In our model, blebbistatin

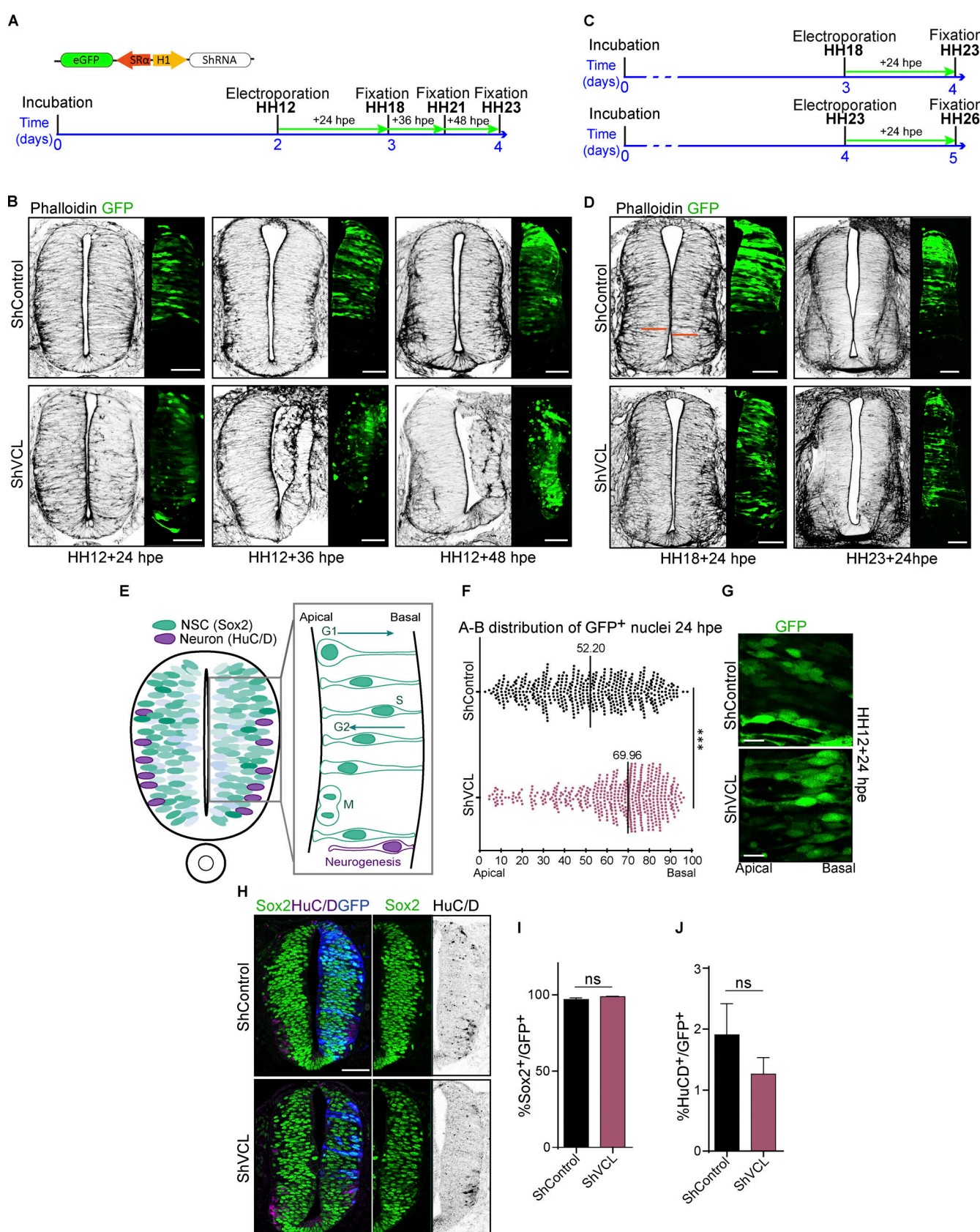

Figure 3. **Vinculin knockdown induces accumulation of NSCs nuclei at the basal side of the neuroepithelium before affecting the actin cytoskeleton structure. (A)** Scheme of the DNA vector and the time schedule used for the experiments shown Fig. 1 B; hpe, hours post-electroporation. **(B)** Images of transverse sections of chicken NTs transfected at stage HH12 with shControl or shVCL for 24, 36, and 48 h stained with phalloidin (actin, gray scale). GFP

indicates transfection (green). Scale bar = 50 µm. **(C)** Time schedules used for the experiments shown Fig. 1 D. **(D)** Images of transverse sections of chicken NTs transfected at stages HH18 or HH23 with shControl or shVCL for 24 h stained with phalloidin (actin, gray scale). GFP indicates transfection (green). Scale bar = 50 µm. **(E)** Drawing illustrating the coordination that exists between INM, cell cycle, and neuron delamination in chicken NT. **(F)** Dot plot showing the apicobasal position of nuclei from NTs transfected at stage HH12 for 24 h with shControl or shVCL. The black bars indicate the median. t test, n = 17 slices from two independent experiments using at least three embryos per condition. **(G)** Representative images from those used to generate the plot shown in 1F. GFP indicates transfection (green). Scale bar = 10 µm. **(H)** Images of NTs transfected at stage HH12 for 24 with shControl or shVCL and stained with antibodies against Sox2 (neural progenitors, green), HuC/D (neurons, magenta). GFP indicates transfection (blue). Sox2 and HuC/D channels are shown separately in green and gray scale, respectively, at the right hand of the panels. Scale bar = 50 µm. **(I and J)** Quantification of the percentage of Sox2+ or HuC/D+ cells relative to GFP cells in the NTs shown in Fig. 3 H. Mean ± SEM. t test, n = 14 sections for control and 13 sections for shVCL, from two independent experiments using at least three embryos per condition. ns = non-significant, *** = P < 0.001.

---

decreased mV but not dV or dR during AB^INM (Fig. 6, A′, B′, and C′; and Table S1) and decreased mV, dV, and dR during BA^INM (Fig. 6, A″, B″, and C″; and Table S1). The fact that the directionality parameters (dV and dR) were already very low in control AB^INM may explain why these two parameters were non-significantly modified by blebbistatin treatment, in spite that a clear decreasing tendency was observed. Therefore, we concluded that INM in chicken NT depends on myosin contraction, especially during BA^INM. On the other hand, in contrast to blebbistatin treatment, vinculin suppression did not modify any of the kinetic parameters during AB^INM (Fig. 6, A′, B′, and C′; and Table S1). However, similarly to blebbistatin, it reduced mV, dV, and dR during BA^INM (Fig. 6, A″, B″, and C″; and Table S1). It has been proposed that actin-dependent AB^INM is a passive movement driven by the compression exerted by the cells doing the BA^INM (Leung et al., 2011). The nuclear movements that we observed in untreated epithelia fit perfectly on this model; besides, the model would explain why blebbistatin treatment affects the BA^INM more than the AB^INM and also why vinculin suppression affects only the BA^INM. Transfected cells are just a small proportion of the cells doing the BA^INM, therefore, even in the case that all transfected cells were halted by vinculin suppression, it would barely reduce the pressure that propels the AB^INM. In contrast, blebbistatin interferes with BA^INM in the whole cell population; consequently, no pressure is generated, and AB^INM is globally inhibited. Notably, the effect of vinculin suppression on INM occurred before any macroscopic disruption of the actin cytoskeleton was observed in the NT. Similarly, we were able to verify that, up to that point, this suppression did not affect the stability of adherens junctions (Fig. S3). Altogether, these results demonstrate that in chicken NT epithelium, INM is driven by actomyosin contraction acting during BA^INM and that vinculin is required for this process.

## NSCs of the NT require vinculin to progress from G2 to M phase

In chicken NT, as in all other pseudostratified epithelia, the cell cycle phases of NSCs are coordinated to INM. Therefore, each phase takes place at specific apicobasal positions of the nucleus. It is now clear that INM requires cell cycle progression (Kosodo et al., 2011; Leung et al., 2011; Strzyz et al., 2015; Ueno et al., 2006); however, controversy remains on whether INM controls the cell cycle. So, we wondered whether cell cycle progression was affected or not by vinculin depletion. Vinculin suppression reduced the number of cells in mitosis (PH3+) already at 24 hpe,

when no changes in cell polarity or epithelium structure were still observed; however, the effect was more dramatic at 36 and 48 hpe (Fig. 7, A and B). Notably, no ectopic mitoses were observed unless the neuroepithelium structure was disrupted. Similar results were obtained by counting the number of mitoses in ex-vivo NT slice cultures, and in this case, blebbistatin treatment reduced the number of mitoses in a similar way to VCL suppression (Fig. 7 C). To find out in which cell-cycle phase the cells were arrested after vinculin suppression, we transfected HH12 chicken NTs with shVCL or shControl for 24 h and added EdU for the last 4 h of incubation to perform a pulse-chase experiment (Fig. 7 D). EdU and mitoses were detected with the fluorescent reagent Click-iT and an anti-PH3 antibody, respectively (Fig. 7 E). Interestingly, we observed that the position of EdU+ nuclei was significantly shifted toward the basal pole in shVCL transfected embryos (Fig. 7 F). In chicken stage HH18 (HH12 + 24 hpe) NTs, the sum of S and G2 phases have a duration of 4–6 h (Kicheva et al., 2014; Molina and Pituello, 2017); consequently, we expected that 4 h after adding the EdU, a good proportion of the cells in mitosis would be EdU+. Interestingly, the percentage was almost 100% in controls, but only 65% in shVCL transfected embryos (Fig. 7 G), in spite that the proportion of transfected cells that incorporated EdU was similar in both groups (Fig. 7 H). These results demonstrated that the absence of vinculin did not affect the S phase, but impeded or greatly delayed the return of the cells to the apical pole during the G2 phase. To confirm this result through a different approach, we cloned the chicken homolog of mAG·hGeminin-1/110 construct (GenBank: AB370333), an S/G2/M fluorescent probe (Sakaue-Sawano et al., 2008). We transfected HH12 chicken NTs for 24 h with shVCL or shControl plus H2B·RFP and ggaGeminin-AG (Fig. 7 I); notably, we observed that suppression of VCL significantly increased the expression of geminin with respect to the controls (Fig. 7 J) and induced a shift toward the basal side of geminin-positive nuclei (Fig. 7 K). In coherence with all these results, cell cycle analysis by flow cytometry showed that the absence of vinculin increased the number of cells in G2/M (from 17% to 31%), reduced them in G0/G1 (from 57% to 26%), and did not change their number in the S (26% and 26%; Fig. 7, L and M) phase. Notably, we observed that epithelial cell lines growing in 2D cultures (HEK-293 or HeLa), which do not present apicobasal polarity and do not perform INM, do not require vinculin for cell cycle progression (Fig. S4). Therefore, we conclude that vinculin is necessary for BA^INM and cell cycle progression from the G2 to M phase in pseudostratified epithelia.

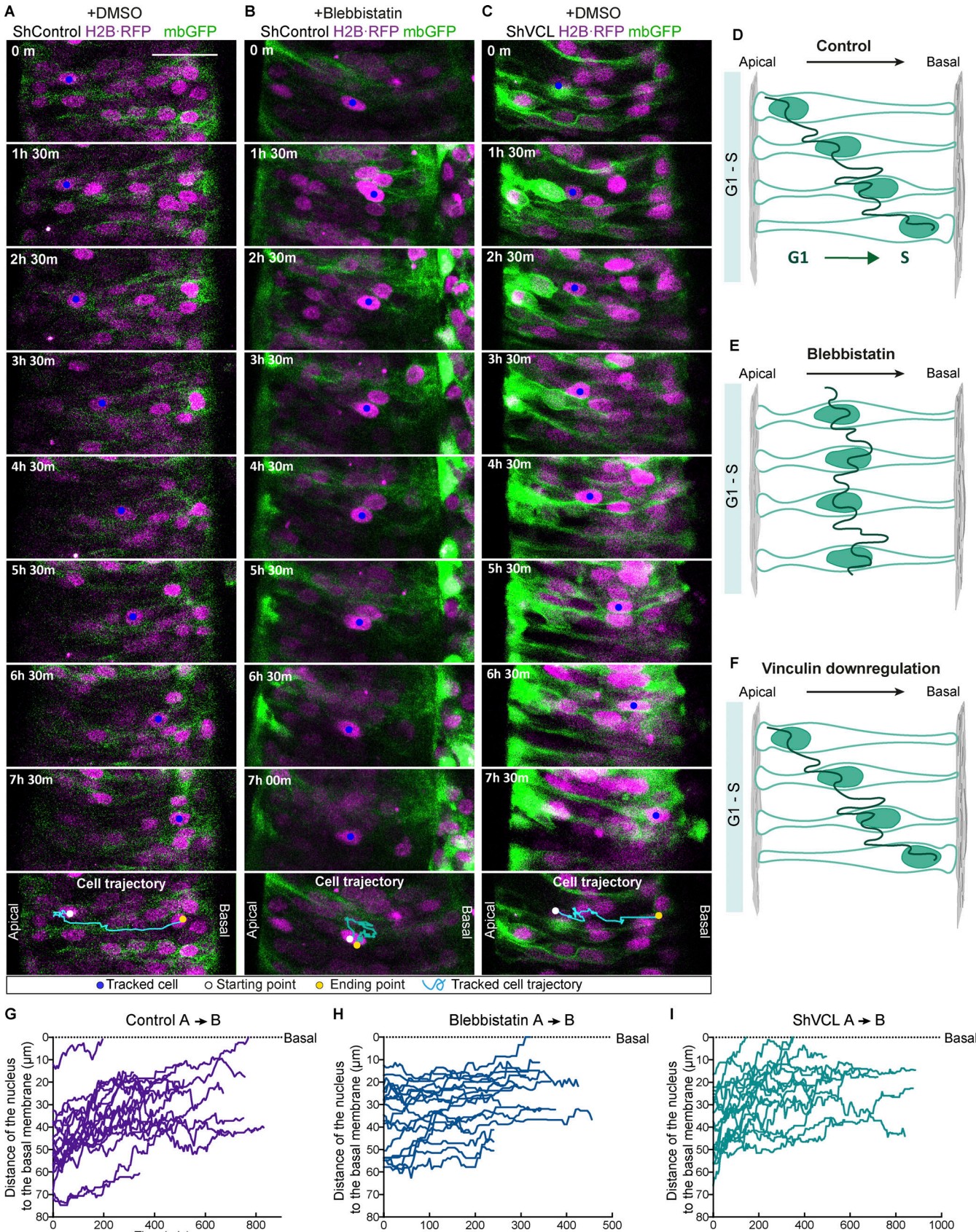

Figure 4. **Apical to basal nuclear migration (AB^INM) in the NT is inhibited by blebbistatin treatment but not by vinculin downregulation. (A–C)** Representative time-series of AB^INM of chicken NTs treated with DMSO and transfected with shControl (A), treated with 12.5 μM of blebbistatin (a myosin II

inhibitor; dissolved in DMSO), and transfected with shControl (B) or treated with DMSO and transfected with shVCL (C). All NTs were also transfected with vectors expressing H2B·RFP (magenta) and membrane-GFP (green) to expose the nucleus and the contour of transfected cells. The blue dot indicates the tracked nucleus in each image. The entire trajectory followed by the tracked nucleus (cyan line), its starting (white dot), and its final (yellow dot) positions are shown in the bottom panel of each treatment. Scale bar = 50 μm. **(D–F)** Drawings simulating the AB$^{INM}$ in each experimental condition: DMSO + shControl (D, control), 12.5 μM blebbistatin + shControl (E, control), or DMSO + shVCL (F, vinculin downregulation). **(G–I)** Plots showing the individual nuclear apicobasal trajectories of 18 cells for each experimental condition, each plot collects cells from two independent experiments: DMSO + shControl (G, control), 12.5 μM blebbistatin + shControl (H, blebbistatin), or DMSO + shVCL (I, shVCL). The X and Y axis represent the elapsed time and the distance to the basal pole, respectively. The basal pole position in the plot is represented by a dotted line.

### Vinculin knockdown impairs centrosome separation and internalization during the G2 phase

The centrosome is a main tubulin-nucleating center that develops essential functions during cell division. In pseudostratified epithelia, such as the NT, centrosomes are located close to the apical surface during G1, S, and part of G2 (Fig. 8 A). During G2, the centrosomes separate, and after that, one of the two centrosomes moves away from the apical membrane internalizing until it reaches the nucleus that is moving in the opposite direction (Spear and Erickson, 2012). The encounter of the centrosome with the nucleus is essential for G2 to M progression (Doxsey et al., 2005), more specifically, centrosome-docked Cdk1–cycB1 initiates the events that lead to mitosis, including the phosphorylation of nuclear heterochromatic histone H3 (Jackman et al., 2003; Krämer et al., 2004). We have shown above that in embryonic NTs, vinculin is mainly distributed at the apical side of neural cell precursors, colocalizing with β-catenin at the AJs and the daughter centriole. Moreover, vinculin suppression inhibits BA$^{INM}$ and the M phase without inducing ectopic mitoses. Consequently, we wondered whether the lack of vinculin was preventing the separation and consequent internalization of the centrosomes during G2/M. Using NTs transfected with Geminin·AG, Cep152·GFP plus Arl13b·RFP and stained with phalloidin, we found that in G2-phase cells, the mother and daughter centrioles were aligned on the apical–basal axis. The mother centriole retained Arl13b staining, while the daughter centriole was internalized below the actin ring plane through an actin-rich environment (Fig. 8 B). Then, we studied the presence of vinculin in centrioles at different stages of internalization in G2-phase cells by transfections of geminin·AG, Cep152·GFP, and vinculin·mCherry. Notably, the maximum vinculin accumulation at the centrosome always coincided with the internalization of the daughter centriole below the AJs plane before moving away from the mother centriole towards the nucleus (Fig. 8 C and Video 8). Interestingly, we observed that after vinculin suppression, the percentage of internalized centrosomes (>2 μm away from the apical membrane) was lower than in control-transfected NTs (Fig. 8, D and E). Moreover, among the internalized centrosomes, the ones from shVCL transfected NTs were closer to the apical membrane than the ones from controls (Fig. 8 F). Therefore, we conclude that vinculin is necessary for centrosome separation and internalization, a process that coordinates BA$^{INM}$ with cell-cycle progression during G2/M phases in pseudostratified epithelia.

### Both the INM and the cell cycle of NT epithelium require the binding of vinculin to β-catenin

Vinculin is an ABP that links the actin cytoskeleton to AJs (cadherin/β-catenin/α-catenin complex) and FAs (integrin/paxilin/talin complex) in cell-to-cell and cell-to-basal membrane adhesion, respectively (Bays and DeMali, 2017). The point mutation A50I in the N-terminal domain of vinculin prevents its interaction with talin and β-catenin, weakening the anchorage of the actin cytoskeleton to adhesion complexes (Peng et al., 2010). We previously demonstrated that in NSCs, vinculin accumulated at the apical domain, associating with β-catenin at the AJs and centrosomes. Therefore, we used the vinculin mutant A50I to study the role of the β-catenin/vinculin interaction on INM and cell cycle. Consistent with our previous findings, open-book preparations revealed that vinculin·mCherry accumulated at the adherens junctions (AJs) and centrioles, which were labeled using an N-cadherin antibody and Cep152-GFP transfection, respectively (Fig. 9 A, arrowheads, and arrows). Interestingly, mCherry or vinculin-A50I·mCherry did not accumulate at all the AJs or centrioles (Fig. 9, A–C). Then, we did rescue experiments to study whether vinculin-A50I could replace endogenous vinculin in INM and cell cycle. Transfection of HH12 chicken embryos with low concentration DNA vectors (0.5 μg/μl) expressing either wild-type vinculin or the A50I mutant did not alter the actin cytoskeleton or change nuclei position at 24 hpe (Fig. 9, E–G and K; control: 47.55%, VCL: 48.69% and VCL$^{A50I}$: 45.5%). In contrast, suppression of vinculin with shVCL shifted the mean nuclei position toward the basal side of the epithelium (Fig. 9, H and K; shVCL: 64.35%). Notably, the basal nuclei accumulation was reverted by wild-type VCL (Fig. 9, I and K; shVCL + VCL: 54.88%) but not by VCL$^{A50I}$ (Fig. 9, J and K; shVCL + VCL$^{A50I}$:60.33%). Similarly, the decrease of mitoses (PH3$^+$) caused by shVCL was rescued by wild-type vinculin, but not by vinculin-A50I (Fig. 9 L). In congruence, cell cycle analysis by flow cytometry showed that the G2/M arrest induced by vinculin suppression was reverted by wild-type vinculin but not by vinculin-A50I (Fig. 9, M–S). The results demonstrate that β-catenin binding to vinculin at the AJs and the centrosome is required in the mechanism by which vinculin drives both BA$^{INM}$ and G2 to M phase progression in the developing NT.

## Discussion

### Role of vinculin at AJs and FAs of the neuroepithelium

In NSCs from NT, actin cytoskeleton spreads from apical to basal membranes and is crucial to maintain the apicobasal polarity, the cell shape, and consequently, the structure of the neuroepithelium. It sticks to apical AJs and basal FAs by interacting with N-cadherin/β-catenin complexes and integrins, respectively. In both cases, the union is supported by a set of ABPs that may vary depending on specific requirements (Arai and Taverna, 2017; Bays and DeMali, 2017). In a recent work, we

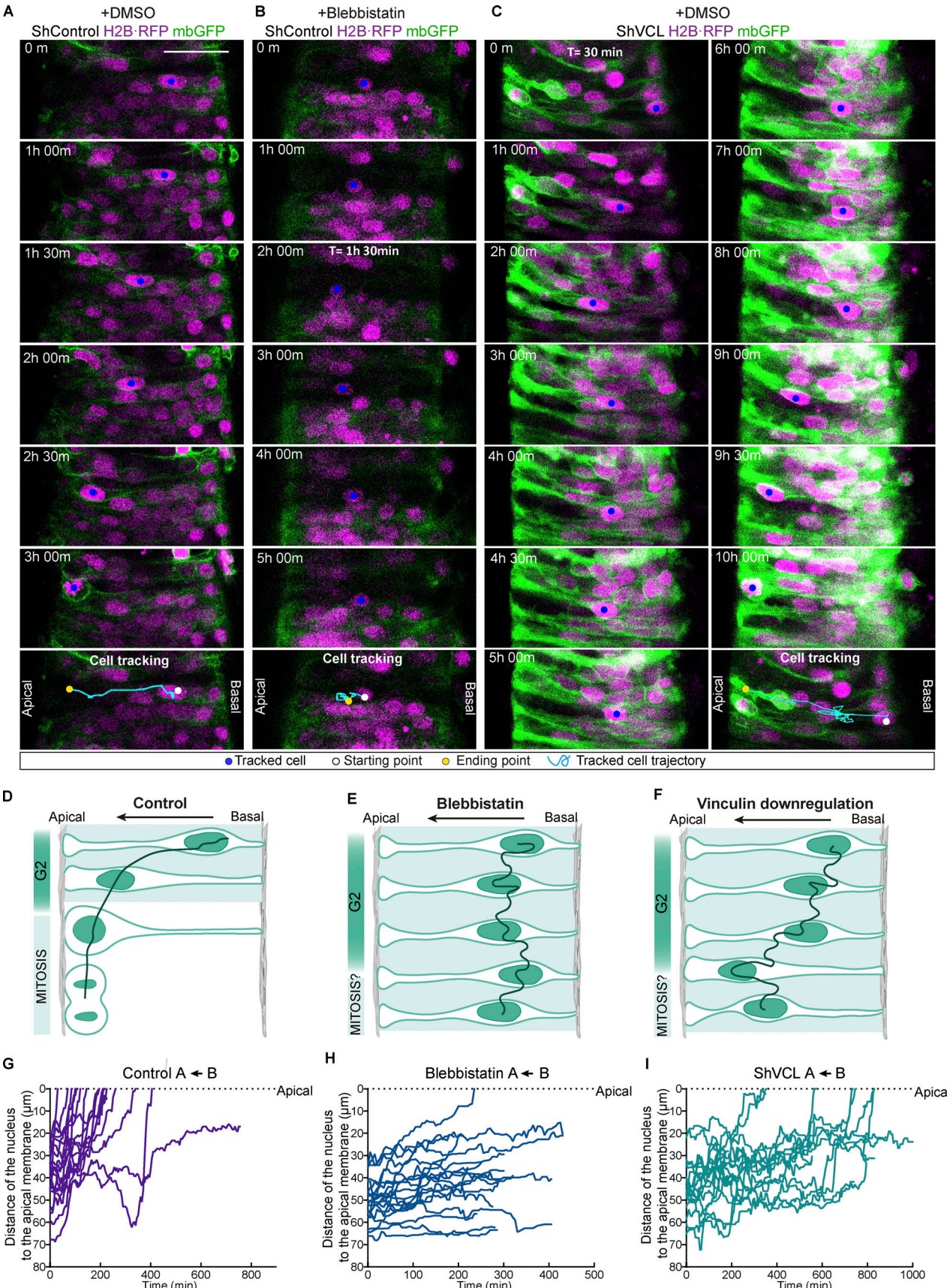

Figure 5. **Basal to apical nuclear migration ($BA^{INM}$) in the NT is inhibited by vinculin downregulation and blebbistatin treatment. (A–C)** Representative time-series of $BA^{INM}$ of chicken NTs treated with DMSO and transfected with shControl (A), treated with 12.5 µM of blebbistatin (a myosin II inhibitor; dissolved

in DMSO) and transfected with shControl (B), or treated with DMSO and transfected with shVCL (C). All NTs were also transfected with vectors expressing H2B·RFP (magenta) and membrane-GFP (green) to expose the nucleus and the contour of transfected cells. The blue dot indicates the tracked nucleus in each image. The entire trajectory followed by the tracked nucleus (cyan line) and its starting (white dot) and final (yellow dot) positions, are shown in the bottom panel of each treatment. Scale bar = 50 µm. **(D–F)** Drawings simulating the BA^INM in each experimental condition: DMSO + shControl (D, control), 12.5 µM blebbistatin + shControl or DMSO + shVCL (E, blebbistatin or vinculin downregulation). **(G–I)** Plots showing the individual nuclear basoapical trajectories of 18 cells for each experimental condition, each plot collects cells from two independent experiments: DMSO + shControl (F, control), 12.5 µM blebbistatin + shControl (G, blebbistatin), or DMSO + shVCL (H, shVCL). The X and Y axis represent the elapsed time and the distance to the apical pole, respectively. The apical pole position in the plot is represented by a dotted line.

identified by mass spectrometry up to six ABPs bound to N-Cadherin via β-catenin in the chicken NT: α-actinin4, α-E-catenin, α-N-catenin, Cortactin, Dbnl, and vinculin (Herrera et al., 2021). Here, we have shown that suppression of vinculin caused major defects in the structure of chicken neuro-epithelium, interestingly, these results agreed with other studies in which suppression of vinculin-related proteins, such as N-cadherin, α-catenin, β-catenin, and Arp2/3 caused adhesion defects, loss of polarity and morphology aberrations in the NSCs of mouse cortex (Chou and Wang, 2016; Kadowaki et al., 2007; Machon et al., 2003; Schmid et al., 2014; Wang et al., 2016). Vinculin has been identified in different cell types as a component of AJs and FAs interacting with α/β-catenin and talin, respectively (Bays and DeMali, 2017), particularly, it has been observed together with the AJs at the apical pole of *Xenopus* neural plate cells (Roffers-Agarwal et al., 2008). We have observed vinculin-mCherrry at both the apical and the basal poles of neural precursors; however, the signal was by far more intense in the apical pole, and its suppression affected principally the apical actin cytoskeleton. Similarly, suppression of vinculin in epithelial cell cultures mostly reduced the vinculin at the AJs, affecting the intercellular adhesions but not the FAs by which the cells attached to the plate (Peng et al., 2010). Cell attachment to basal membranes or culture dishes is more stable than cell-to-cell interactions; congruently, vinculin binds with greater affinity to FAs than to AJs (Bakolitsa et al., 2004). Stable structures need less molecular recycling than dynamic ones. Therefore, the fact that after vinculin suppression, we primarily observed defects in the AJs, does not exclude the possibility that vinculin is equally required to stabilize the actin cytoskeleton at the basal FAs; on the contrary, it could indicate different dynamics of the two vinculin pools.

## INM in chicken NT

During INM, nuclei of proliferating NSCs occupy different positions along the apicobasal axis throughout different cell cycle phases. In G1/S phases, nuclei migrate from apical to basal in a passive and stochastic way (Kosodo et al., 2011; Leung et al., 2011; Norden et al., 2009). In contrast, during G2/M phases, nuclei migrate from basal to apical by a directed active process; however, the entity of the cytoskeleton supporting this apically directed movement prior to mitosis remains controversial. It has been proposed that depending on the type of neuroepithelium, BA^INM may be either actin or tubulin-dependent (Norden, 2017). Some studies have demonstrated that in longer neuroepithelia, such as the murine cerebral cortex, BA^INM is driven by the tubulin cytoskeleton (Hu et al., 2013; Schenk et al., 2009; Spear and Erickson, 2012; Tsai et al., 2010). In contrast, other studies

done in shorter neuroepithelia, such as the ones found in the zebrafish hindbrain and retina, show that it is the actin cytoskeleton that carries out this movement (Leung et al., 2011; Norden et al., 2009; Strzyz et al., 2015; Yanakieva et al., 2019). The neuroepithelium of chicken NT has morphology, actin distribution, and apicobasal length similar to those of zebrafish hindbrain (Norden, 2017; Yanakieva et al., 2019). However, BA^INM was reported to depend on actomyosin in zebrafish hindbrain (Yanakieva et al., 2019) and on tubulin in chicken NT (Spear and Erickson, 2012). Spear's work used cytochalasin B, an inhibitor of actin polymerization, to rule out the implication of actomyosin on BA^INM. In this study, the chicken NSCs treated with cytochalasin B completed the G2 phase, internalized the centrosome, and initiated the breakdown of the nuclear envelope and the mitosis; however, they did not adopt the characteristic rounded shape of NSCs in mitosis. Besides, another study inhibited actin nucleation, which is the first step of actin polymerization, in zebrafish hindbrain by using different drugs or overexpressing a dominant-negative Arp2/3. Notably, it had no effect on BA^INM, demonstrating that, as in chicken NT, this movement does not require actin polymerization (Yanakieva et al., 2019). Here, we have shown that in chicken NT, blebbistatin, and vinculin suppression impair BA^INM, centrosome internalization, and mitosis initiation. We used a blebbistatin concentration that allows for inhibiting myosin II contraction without affecting actin polymerization (Straight et al., 2003). Besides, vinculin reinforces the anchorage of the actin cytoskeleton in AJs and FAs also through a myosin II-dependent mechanism (Ciobanasu et al., 2014; Huang et al., 2017; le Duc et al., 2010). Therefore, in spite of the fact that actin polymerization may not be required, it cannot be denied that actomyosin cytoskeleton participates in the BA^INM of chicken NT. As proposed for other epithelia, actin and tubulin cytoskeletons might work in coordination during BA^INM in chicken NT (Meyer et al., 2011). In fact, at the apical pole of chicken NSCs, the centrosome nucleates microtubules that are directed towards the nucleus, while others align with the apical actin belt, forming a wheel-like configuration where the actin and tubulin cytoskeletons are interconnected (Kasioulis et al., 2017). The actin belt binds to the apical AJs by a mechanism involving β-catenin and ABPs. Here, we have shown the significance of vinculin, one of these ABPs, in strengthening the binding between the actin belt and the apical AJs. Therefore, the reinforcement provided by vinculin might play a crucial role when the microtubules anchored in the centrosome exert forces to move the nucleus toward the apical pole during BA^INM. To summarize, considering the collective evidence, we propose an integrated model where vinculin enables both the actin and tubulin cytoskeletons to withstand the

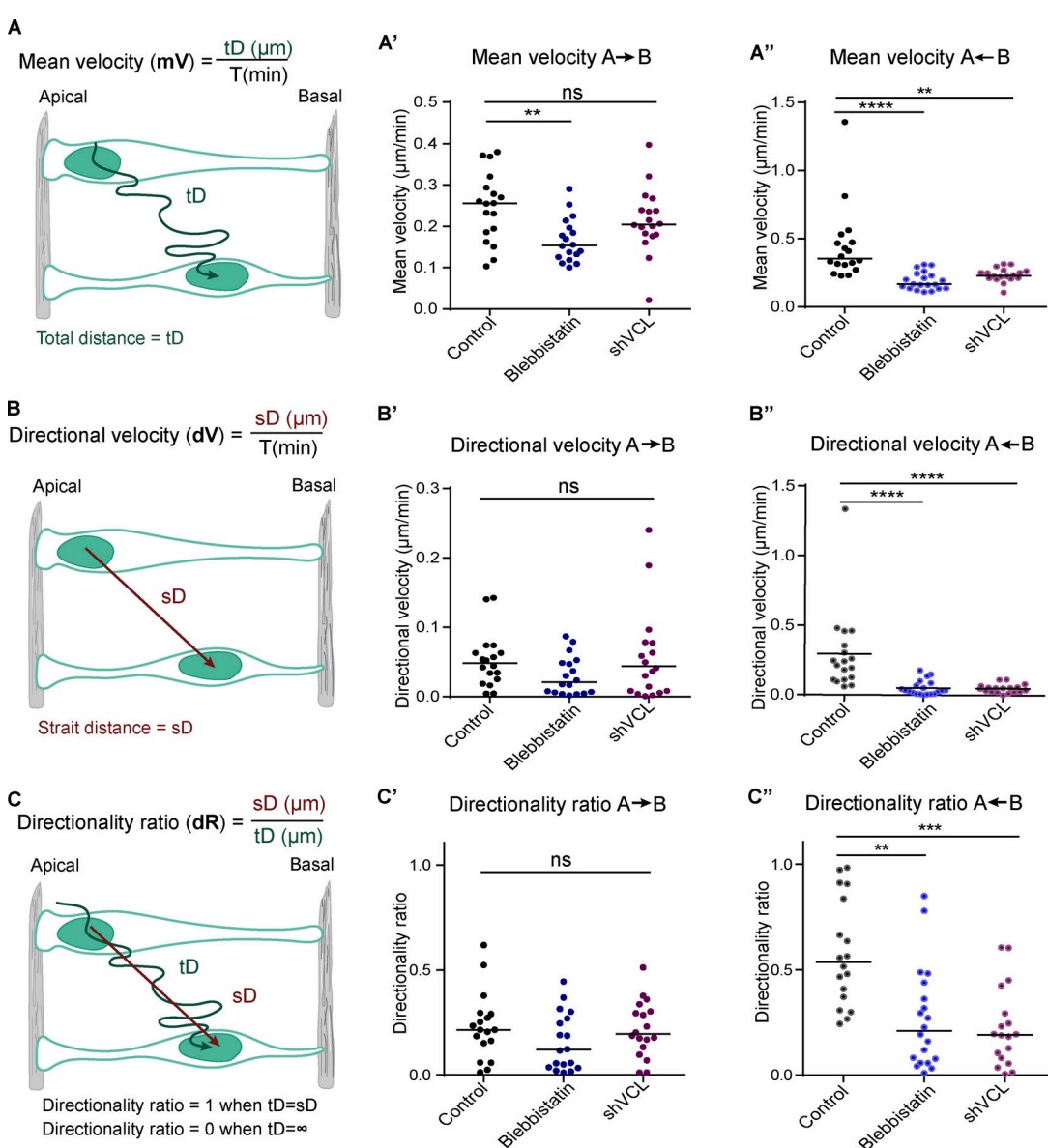

Figure 6. **Vinculin controls velocity and directionality during basal to apical nuclear migration in the NT. (A–C)** Formulas used and explicative schematic representations of the kinetic parameters studied from apicobasal and basoapical movements in NTs treated with blebbistatin or transfected with shVCL: mean velocity (mV, A), directional velocity (dV, B), and directionality ratio (dR, C). **(A'–C')** Dot plots showing the apicobasal mean velocity (A'), directional velocity (B'), and directionality ratio (C') of NTs treated with shControl + DMSO (control), 12.5 μM blebbistatin + shControl (blebbistatin) or DMSO + shVCL (shVCL). **(A''–C'')** Dot plots showing the baso-apical mean velocity (A''), directional velocity (B''), and directionality ratio (C'') of NTs treated with shControl + DMSO (control), 12.5 μM blebbistatin + shControl (blebbistatin), or DMSO + shVCL (shVCL). tD = total distance; sD = strait distance; T = time in minutes. ∞ = infinite. The black bars indicate the mean. One-way ANOVA plus Dunn's multi-comparisons test, $n$ = 18 cells per condition collected from two independent experiments using at least three embryos per condition. ns: non-significant, ** = $P < 0.01$, *** = $P < 0.001$, **** = $P < 0.0001$.

mechanical forces required for the BA[INM] in the neuro-epithelium of chicken NT.

**Is the centrosome the link between deficient INM and cell cycle arrest after vinculin suppression?**

In this study, we show that the vinculin/β-catenin complex accumulates at the daughter centriole during centrosome separation, a process that takes place just before mitosis. Besides, suppression of vinculin causes defects in centrosome separation and internalization, inhibits BA[INM], and arrests the cell cycle at the G2 phase. Additionally, both BA[INM] inhibition and G2 arrest

require vinculin/β-catenin interaction at the AJs and the centrosome. Interestingly, G2 arrest induced by the suppression of vinculin significantly decreased the number of apical mitoses without inducing ectopic ones. In neuroepithelial cells, mitoses always culminate at the apical surface (Taverna and Huttner, 2010), but this process is initiated slightly earlier at the proximity of the apical pole when the internalized centrosome reaches the nucleus (Spear and Erickson, 2012). Although in our study the internalization of the centrosomes is affected, it has been reported that INM is independent of the initial centrosome position in zebrafish retinal neuroepithelium (Strzyz et al.,

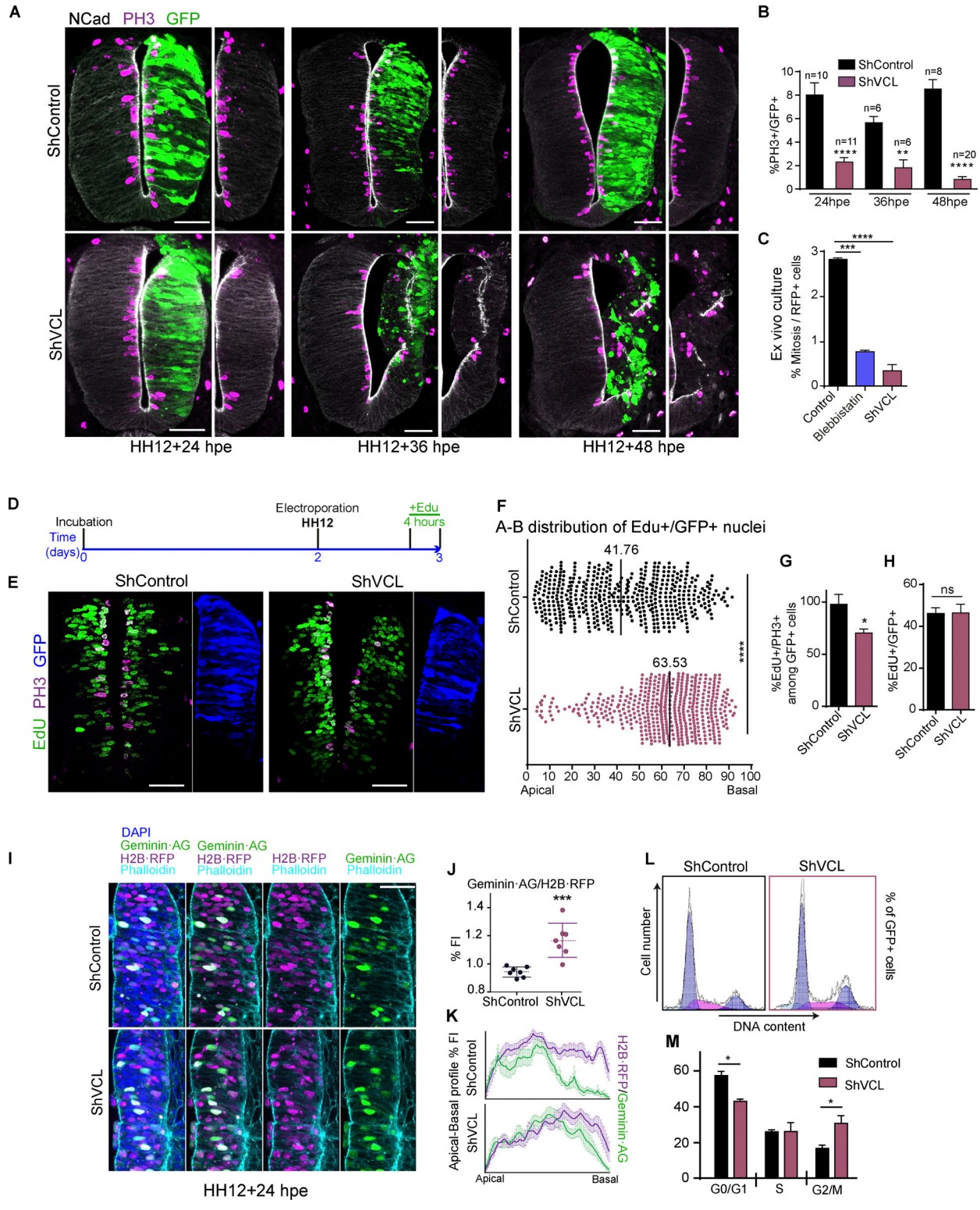

Figure 7.  **NSCs from NT require vinculin to progress from G2 to M phase. (A)** Images of transverse sections of chicken NTs transfected at stage HH12 for 24, 36, and 48 h with shControl or shVCL, stained with anti N-cadherin (grayscale) and anti PH3 (magenta), GFP (green) indicates transfection. The transfected halves of the NTs are shown without the GFP channel at the right hand of each image for clarity. Scale bar = 50 μm. **(B)** Bar graph showing the percentage of mitotic cells (PH3⁺) relative to GFP population in NTs transfected with shControl or shVCL for 24, 36, and 48 h. Mean ± SEM. One-way ANOVA plus Tukey's multi-comparisons test, *n* = 6–20 sections per condition from three experiments. **(C)** Bar graph showing the percentage of mitosis relative to transfected cells in ex-vivo chicken NT slices from Fig. 5; cultured in control, blebbistatin, or shVCL conditions. Mean ± SEM. One-way ANOVA plus Tukey's multi-comparisons test, *n* = 29,072 cells for control, 22,087 for blebbistatin and 18,951 for shVCL, from two movies. **(D)** Time schedule used for the experiments shown in Fig. 7, E–H. **(E)** Images of transverse sections of chick NTs electroporated at stage HH12 for 24 h with shControl or shVCL, the embryos were treated in ovo with EdU for the last 4 h before fixation. The sections were stained with anti-PH3 (magenta) and anti-EdU (green). GFP (blue) shows transfection. Scale bar = 50 μm.

**(F)** Dot plot showing the apicobasal position of EdU⁺/GFP⁺ nuclei from NTs transfected at stage HH12 for 24 h with shControl or shVCL. The black bars indicate the median. *t* test, *n* = 8 sections for control and 11 sections for shVCL, from two independent experiments using at least three embryos per condition. **(G)** Bar graphs showing the percentage of mitotic cells (PH3⁺) that had incorporated EdU among the GFP population in NTs transfected at stage HH12 for 24 h with shControl or shVCL. Mean ± SEM. *t* test, *n* = 15 sections for control and 15 for shVCL from three independent experiments using at least three embryos per condition. **(H)** As in G but counting all the cells that incorporated EdU among the GFP population. Mean ± SEM. *t* test, *n* = 15 sections for control and 16 for shVCL from three experiments using at least three embryos per condition. **(I)** Images of transverse sections of chicken NTs transfected at stage HH12 for 24 h with geminin·AG (green), H2B·RFP (magenta), and pSUPER-shControl or pSUPER-shVCL (vectors without GFP), stained with phalloidin (cyan) and DAPI (blue). Scale bar = 50 µm. **(J)** Dot plot showing the relative FI of Geminin·AG versus H2B·RFP in ShControl or ShVCL transfected NTs. Mean ± SEM. *t* test, *n* = 7 slices from two experiments using at least three embryos per condition. **(K)** Plot profiles showing the FI of Geminin·AG (green) and H2B·RFP (purple) along the apicobasal axis 24 hpe of ShControl or ShVCL. Mean ± SD. *n* = 7 sections for control and 7 for shVCL from two experiments using at least three embryos per condition. **(L)** Flow cytometry analysis profiles showing DNA content (Hoechst fluorescence intensity) and cell number of GFP⁺ population from chicken NTs transfected at stage HH12 with shControl or shVCL for 24 h **(M)** Bar graph showing the percentage of cells in each cell cycle phase calculated using the profiles shown in Fig. 7 L. Mean ± SEM. *t* test. Two experiments using at least three embryos per condition. A minimum of 2,000 cells were analyzed in each experiment. ns = non-significant, * = P < 0.05, ** = P < 0.01, *** = P < 0.001, **** = P < 0.0001.

2015). Even when the centrosomes are non-apically associated, nuclei move to the apical surface to perform mitosis. Conversely, ectopic mitoses have been observed in models in which INM has been studied using inhibitors of actin polymerization. In *Drosophila* imaginal discs, treatment with latrunculin A or cytochalasin D inhibited INM and induced the presence of ectopic mitoses out of the mantle zone plane (Meyer et al., 2011). In addition, Spear and Erickson observed ectopic mitoses near the apical pole in HH12-16 chicken NTs treated with cytochalasin B, although in this case, neither BA^INM nor centrosome was affected (Spear and Erickson, 2012). On the other hand, the presence of ectopic mitoses when INM is artificially halted by the inhibition of myosin-II remains controversial. Experiments conducted on zebrafish retina detected ectopic mitoses in fixed-tissue samples treated with blebbistatin at 200 µM (Strzyz et al., 2015). However, another study using in vivo time-lapse also on zebrafish retina reported the inhibition of INM but not the presence of apical or ectopic mitoses in embryos treated with 100 µM blebbistatin for over 10 h (Norden et al., 2009). Similarly, we here demonstrate that either vinculin suppression or blebbistatin treatment stops both INM and cell cycle in chicken NT, but in neither case, ectopic mitoses are observed. Although in the previous study, a higher concentration of blebbistatin (100 µM) than in ours (12.5 µM) was used, this concentration was still half of that used in Strzyz et al. (2015). It is worth noting here that blebbistatin is a myosin II inhibitor that, used at low doses, interferes with actomyosin contraction but does not affect the epithelium structure (Schenk et al., 2009). Therefore, the possibility of off-target effects at high concentrations cannot be excluded. In any case, the progression of the cell cycle from G2 to M, whether at the apical membrane proximity or at ectopic positions, necessitates the separation of the centrosomes to move away from each other to establish the mitotic spindle (Doxsey et al., 2005; Jackman et al., 2003; Krämer et al., 2004). Besides, evidence exists that cell cycle arrest consistently stops INM (Kosodo et al., 2011; Leung et al., 2011; Strzyz et al., 2015; Ueno et al., 2006). Notably, in these studies where the authors have used different methods to inhibit the progression of the cell cycle, and similarly to what we have observed by suppressing vinculin or treating with blebbistatin, the cell cycle arrest has always been associated with a halt in nuclear interkinetic movement. Therefore, the concurrence of INM inhibition and cell cycle arrest shall depend on the role of vinculin in centrosome separation and internalization to allow mitotic spindle formation; however, we

must admit that with the tools and data presented, a direct effect on the FAs of the basal pole cannot be definitively ruled out, which could indirectly interfere with INM or centriole internalization.

## Materials and methods
### Commercial antibodies and chemicals
Mouse antibodies against HuC/D (1:500, #A21271; Molecular Probes), Strep-tag (1:500, #2-1507-001; IBA), and Vinculin (1:5,000; #V9131; Sigma-Aldrich).

Rat antibodies against N-Cadherin (1:200, #13-2100; Invitrogen) and PH3 (1:1,000, #06-570; Millipore).

Rabbit antibodies against Sox2 (1:500, #48-1400; Invitrogen).

Chemicals: Rhodamine-Phalloidin (1:250, #R415; Invitrogen), Blebbistatin (#B0560-1MG; Sigma-Aldrich), and EdU (#C10340; Life technologies).

### DNA constructs
Proteins were expressed using pCIG, pEGFP-C1, pmCherry-C1, pCAGGS, or pCS2 vectors and short hairpin inhibitory RNAs were generated with pSHIN (a vector with bicistronic GFP as reporter) or pSUPER (same vector without GFP). Substitution mutants were generated through standard techniques and all constructs were sequenced prior to use.

#### *pCIG constructs*
- VCL-WT: coding sequence (CDS) of mouse vinculin subcloned from pEGFP-C2-VCL-WT, which was kindly provided by Dr. Wolfgang Ziegler. Generated in this study.
- VCL-A50I: mouse vinculin mutant in which the residue A50 was mutated to isoleucine using pCIG-VCL-WT as a template. Generated in this study.
- β-Catenin·ST: ST tagged Human β-Catenin^S33Y. To generate ST-tagged β-Catenin, three copies of the *Strep*-tag II peptide (WSHPQFEK) were cloned at the C-terminus of human β-Catenin^S33Y CDS. Generated in our lab (Herrera et al., 2021).

#### *pCIEGO constructs*
- ggGeminin-AG: Amino acids 75–190 of X1 transcription variant of chicken geminin fused to the C-terminus of azami green protein. Generated in this study.
- β-Catenin·FLAG: Human β-Catenin^S33Y with a FLAG sequence (DYKDDDDK) cloned at the N-terminus. Generated in our lab (Herrera et al., 2014).

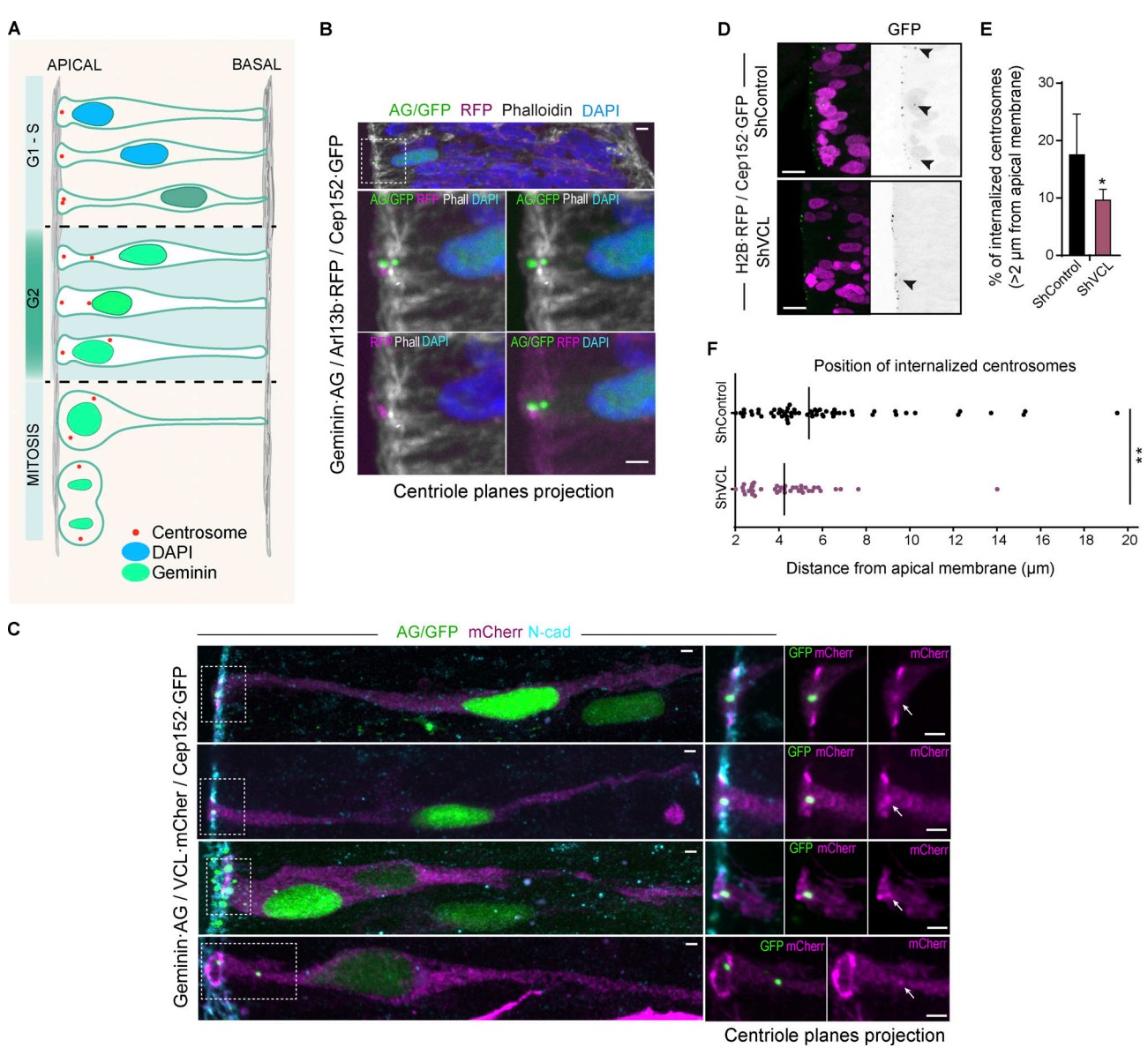

Figure 8. **Interaction of vinculin with adhesion complexes is required for its functions in INM and cell cycle. (A)** Drawing showing an idealization of the positions occupied by the nucleus and the centrosomes along INM and cell cycle. **(B)** Images of transverse sections of chicken NTs transfected at stage HH12 for 24 h with geminin·AG (green), Arl13b·RFP (cilia marker, magenta), and Cep152·GFP (centriole marker, green) and stained with phalloidin (grays) and DAPI (blue). The area encircled in the top panel is magnified in the lower panels, in which different combinations of channels are displayed for clarity. Scale bar = 2 μm. **(C)** Images of transverse sections of chicken NTs transfected at stage HH12 for 24 h with geminin·AG (green), VCL·mCherry (magenta), and Cep152·GFP (green) and stained with N-cadherin (cyan). Four different geminin·AG+ cells with the nucleus at different apical-basal positions are shown. The areas encircled in the left panels are magnified in the right panels, in which different combinations of channels are displayed for clarity. Arrows indicate the centriole position in the VCL·mCherry channel. Scale bar = 10 μm. **(D)** Apical region images of NTs electroporated at stage HH18 for 24 h with H2B-RFP (magenta), Cep152-GFP (green or gray), and shControl or shVCL. Arrowheads point to internalized centrosomes. Scale bar = 10 μm. **(E)** Bar graph showing the percentage of internalized centrosomes (>2 μm from the apical membrane) in NTs transfection with shControl or shVCL. Mean ± SEM. $t$ test, $n$ = 15 sections for control and 10 for shVCL from three experiments using at least three embryos per condition. **(F)** Dot plot showing the distance from the apical membrane to the centrosome of all internalized centrosomes (>2 μm from the apical membrane) from the data shown in Fig. 8 F. The black bars indicate the median. $t$ test. * = P < 0.05, ** = P < 0.01.

### pEGFP-C1 constructs
- Cep152-GFP: provided by Dr. Elisa Martí. (Saade et al., 2020).
- Arl13b-GFP: provide by Dr. Judith Paridaen (Paridaen et al., 2013).

### pCAG-2A constructs
- Arl13b·RFP: provided by Dr. Elisa Martí (Saade et al., 2020).

### pmCherry-C1 constructs
- VCL·mCherry: chicken vinculin fused to the fluorescent protein mCherry. Generated in this study.
- VCL·mCherry[A50I]: chicken vinculin A50I fused to the fluorescent protein mCherry. Generated in this study.
- VCL[T12]·mCherry: constitutively active form of chicken vinculin with the residues D974, K975, R976, and R978 mutated to

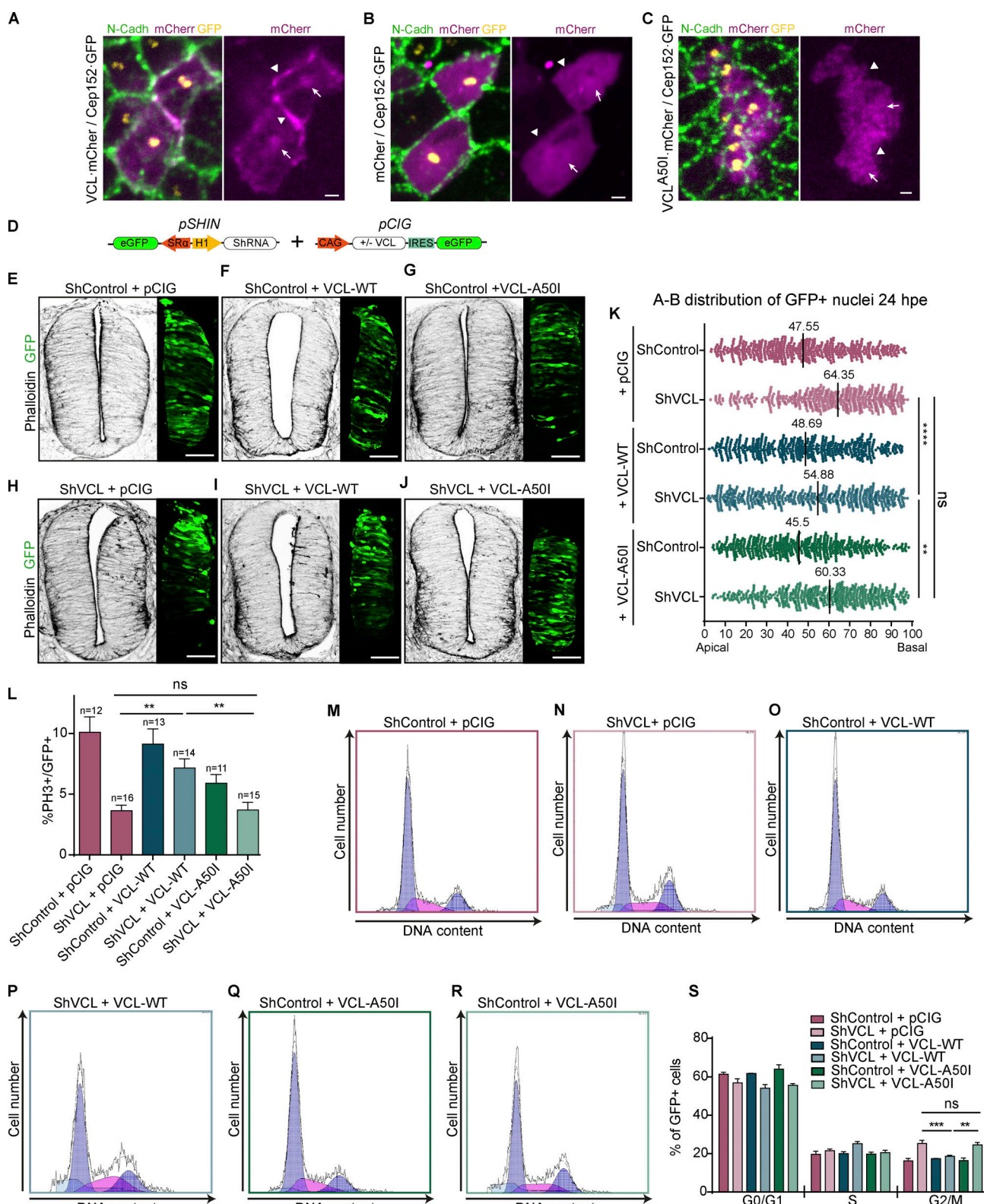

Figure 9. **β-catenin targets vinculin to AJs and centrioles. (A–C)** "En face" pictures of chicken NTs transfected at stage HH12 for 24 h with Cep152-GFP (yellow) and VCL·mCherry (A) or mCherry (B) or VCL$^{A50I}$·mCherry (magenta) and stained with anti N-cadherin antibody (green). The magenta channel is shown alone in right-handed panels for clarity. The position of representative AJs and centrosomes are indicated by arrowheads and arrows, respectively. The images are Z projections of the confocal planes containing the AJs and the centrioles (7 to 10 planes spaced 0.15 μm). Scale bar = 2 μm. **(D)** Scheme of the DNA vector used in this figure. **(E–J)** Images of transverse sections of chicken NTs transfected at stage HH12 for 24 h with shControl or shVCL plus empty vector (pCIG), VCL-WT, or VCL-A50I, stained with phalloidin (actin, gray scale), the GFP expressed in the transfected halves of the NTs is shown at the right hand of each image to show transfection level. Scale bar = 50 μm. **(K)** Dot plot showing the apicobasal position of the nuclei from the NTs shown in Fig. 9 E. The apical–basal

position of 500 nuclei was measured for each condition. The black bars indicate the median. One-way ANOVA plus Dunn's multiple comparisons test. $n = 10$ sections per condition from three experiments using at least three embryos per condition. **(L)** Bar graph showing the percentage of mitotic (PH3+) cells relative to transfected cells (GFP+) in chicken NTs electroporated at stage HH12 for 24 h with shControl or shVCL plus empty vector (pCIG), VCL-WT, or VCL-A50I. Mean ± SEM. One-way ANOVA plus Tukey's multicomparison test, $n = 11$–16 sections from two experiments using at least three embryos per condition. **(M–R)** Flow cytometry analysis profiles showing DNA content (Hoechst fluorescence intensity) and cell number of GFP+ population from chicken NTs transfected at stage HH12 for 24 h with shControl or shVCL plus empty vector (pCIG), VCL-WT, or VCL-A50I. **(S)** Bar graph showing the percentage of cells in each cell cycle phase from the profiles shown in Fig. 9, M–R. Mean ± SEM. Two-way ANOVA plus Tukey's multicomparison test. Two experiments using at least three embryos per condition. A minimum of 2,000 cells per condition were analyzed in each experiment. ns = non-significant, * = $P < 0.05$, ** = $P < 0.01$, *** = $P < 0.001$, **** = $P < 0.0001$.

---

alanine. These mutations interrupt the head–tail interaction of vinculin facilitating its binding to actin (Cohen et al., 2005). In this vector, vinculin-T12 is fused to the fluorescent protein mCherry and was used for vinculin localization studies. Provided by the laboratory of Dr. Nicolás Borghi (Liu et al., 2016).

### pCAGGS constructs
- H2B·RFP: expression vector encoding histone 2B fused to RFP. Provided by the laboratory of Dr. Mariane Bronner (Strobl-Mazzulla et al., 2010).

### pCS2 construct:
- mb-GFP, green fluorescent protein with a glyco-sylphosphatidylinositol (GPI) anchor motif fused to the C-terminus. Provided by the laboratory of Dr. Kate Storey (Das and Storey, 2012).

### pSHIN constructs to produce
- ShRNAi targeting the following chicken genes: ACTN4, sh1 (5′-CCCGACGAGAAGGCCATCA-3′), sh2 (5′-ATCAAACTGTCCGGG AACA-3′); CTNNA1, sh1 (5′-AGAAGAAGCACGTCAACCC-3′), sh2 (5′-ATGGCCTCGCTCAACCTCC-3′); CTNNA2, sh1 (5′-ATG GCAGACTCCTCGTGCA-3′), sh2 (5′-AGTAGCACAGGACAACAT G-3′); CTTN, sh1 (5′-AAGAACAGGAAGACCGAAG-3′), sh2 (5′-ACAAAGAATGGCACGGGAG-3′); DBNL, sh1 (5′-GAAATCAAG CGGGTCAACA-3′), sh2 (5′-GAATCAGTGGAGAAAGCCC-3′); VCL, sh1 (5′-GCAGACAACAGAAGACCAG-3′), sh2 (5′-GGAATG ACAAAGATGGCGA-3′); shControl, (5′-CCGGTCTCGACGGTC GAGT-3′). Generated in this study.
- ShRNAi targeting the following human genes: VCL, sh1 (5′-TCAAACCACTGAGGATCAG-3′), sh2 (5′-GCAGAATTATGT GATGATC-3′); shControl, (5′-CCGGTCTCGACGGTCGAGT-3′). Generated in this study.

### pSUPER constructs to produce
- ShRNAi targeting the following chicken genes: VCL, sh1 (5′-GCAGACAACAGAAGACCAG-3′), sh2 (5′-GGAATGACAAAG ATGGCGA-3′), and shControl, (5′-CCGGTCTCGACGGTCGAGT-3′). Generated in this study.

### Chick embryo in ovo electroporation
Eggs from White-Leghorn chickens were incubated at 37.5°C in an atmosphere of 45% humidity and the embryos were staged according to Hamburger and Hamilton (HH; Hamburger and Hamilton, 1992). Chick embryos were electroporated with column-purified plasmid DNA (1 µg/µl for pCIG vectors, 3 µg/µl for pSHIN and pSUPER vectors, and 0.2 µg/µl for the rest of

vectors) in $H_2O$ containing Fast Green (0.5 µg/µl). Briefly, plasmid DNA was injected into the lumen of HH12, HH18, or HH23 NTs, electrodes were placed on either side of the NT and electroporation was carried out by applying five 50-ms square pulses using an Intracel Dual Pulse (TSS10) electroporator set at 25 V. Transfected embryos were allowed to develop to the specific stages and then dissected under a fluorescence dissection microscope. In our conditions, HH12 embryos electroporated for 24, 36, and 48 h normally reached stages HH18, HH21, and HH23 respectively; HH18 embryos electroporated for 24 h reached stage HH23; and HH23 embryos electroporated for 24 h reached stage HH26. Embryos that did not develop to the expected stages were discarded. To stage the different embryos we used the staging guide published in: https://embryology.med.unsw.edu. au/embryology/index.php/Hamburger_Hamilton_Stages.

### Culture and electroporation of cells
HEK-293T (#CRL-3216; ATCC) and HeLa (#CCL-2; ATCC) cell lines were cultured in DMEM/F12 and DMEM (high glucose pyruvate), respectively, supplemented with 10% fetal bovine serum (FBS) and 50 mg/l of penicillin-streptomycin. CEFS were obtained from chicken embryos at stage HH30-35 (7–9 d of development). The embryos were mechanically disaggregated with forceps and then treated with trypsin-EDTA for 15 min at 37°C, DNAse I (final concentration 100 U/ml) was added to digest the DNA resulting from cell damage, trypsin was inactivated by adding one volume of culture medium with 10% FBS. Cell suspension was centrifuged for 5 min at 300 $g$ to eliminate aggregates and the supernatant was resuspended in Optimen at the desired cell concentration. For electroporation, CEFs were resuspended at 15,000,000 cells/ml. Cell cultures were maintained in an atmosphere of 5% $CO_2$ at a temperature of 37°C.

Transient transfection of cell lines and CEFs was carried out with an electroporator (Microporator MP 100; Digital Bio) applying a pulse of 1,200–1,400 V for 20 ms. Transfected cells were seeded according to the experiment to be carried out. For real-time PCR (RT-qPCR) experiments, 1.5–3 million cells per well were seeded in six-well culture plates and grown for 24 h. For immunostaining, 0.5–1 million cells per well were seeded in 24-well culture plates on coverslips. Cultures reached 70–80% confluence at harvest time.

### Purification of Strep-tag II tagged proteins on Strep-Tactin columns
We adapted the protocols provided by the manufacturer to chicken-embryo model to purify proteins carrying ST tags and their interacting partners with *Strep*-Tactin affinity columns

(Herrera et al., 2021). *Strep*-tag II is a short peptide (eight amino acids, WSHPQFEK) that binds with high affinity/selectivity to *Strep*-Tactin, an engineered streptavidin. For purification experiments, chicken embryos were electroporated for 24 h with tagged β-Catenin, GFP⁺ areas of the NTs dissected out in cold PBS (137 mM NaCl, 2.7 mM KCl, 10 mM $Na_2HPO_4$, 1.8 mM $KH_2PO_4$ [pH 7.4]) and dissolved in PIK buffer (20 mM Tris-HCl [pH 7.4], 137 mM NaCl, 10% Glycerol, 1% NP-40, 1 mM $CaCl_2$, 1 mM $MgCl_2$, 10 μg/ml Aprotinin, 10 μg/ml Leupeptin and 1 mM PMSF). 1 ml of PIK buffer was used for every 15 embryos (HH18 embryos). The insoluble material was removed by centrifugation at 18,500 × *g* for 10 min. Purification was performed on prepacked *Strep*-Tactin (0.2 ml bed volume [bv] gravity-SuperFlow columns: #2-1209-550; IBA Life-sciences) following the manufacturer's instructions and using the reagents provided with the columns. Briefly, columns were pre-stabilized with 5 bvs of lysis buffer and 1 ml of lysate was applied to each 0.2 ml column, washed with 5–15 bvs, and finally eluted in 100 μl aliquots. Purified proteins were stored at –80 for further use.

### Immunoblotting
Purified fractions were mixed with 5× SDS Laemmli sample buffer (1× = 2% SDS, 65 mM Tris-HCl [pH 6.8], 10% Glycerol, 100 mM DTT, and 0.5 mg/ml Bromophenol Blue), boiled for 5 min, resolved by SDS-PAGE (8% gels), and transferred to ni-trocellulose membranes. The membranes were blocked with 1:1 TBS (150 mM NaCl and 20 mM Tris-HCl [pH 7.4]) and Odyssey Blocking Buffer (OBB, Li-COR, inc.), incubated with the primary antibodies in OBB containing 0.2% of Tween-20, washed three times with TTBS (150 mM NaCl, 0.1% Tween-20 and 20 mM Tris-HCl [pH 7.4]) and incubated with Alexa-labeled secondary antibodies in OBB containing 0.2% of Tween-20 and 0.01% of SDS. After three final washes in TTBS, the membranes were allowed to dry, and the fluorescence was detected on the Odyssey Infrared Imaging System (LI-COR). The molecular weights were calculated using Bio-Rad Precision Molecular Weight Markers.

### Immunostaining
Embryos were fixed overnight at 4°C in 4% PFA (4% parafor-maldehyde in PBS) and then sectioned at 60 μm thickness with a Vibratome (VT1000S; Leica) or opened by sectioning along the roof plate for open book preparations. HEK and HeLa cells were fixed in 4% PFA at RT for 10 min. Immunostaining was performed following standard procedures. After washing in PBT (PBS containing 0.1% of Triton X-100), samples were incubated with the appropriate primary antibodies and developed with Alexa or Cy-anine conjugated secondary antibodies. After staining, the sections or the open book preparations were mounted with Fluoromount (Sigma-Aldrich) and examined at 18°C on a Leica SP5 (20× NA 0.7, 40× NA 1.25) controlled under the Leica LAS software or a Zeiss Lsm 780 (25× NA0.57, 40× NA 1.3, 63× NA0.18) multiphoton microscope controlled under the LSM Software ZEN 2.1 or a Drag-onfly 500 (Oxford Instruments) using a 100× objective (NA 2.1). Images were manipulated using ImageJ software.

### In situ hybridization
HH18 chick embryos were fixed overnight at 4°C with 4% PFA. Next day, embryos were dehydrated with increasing concentrations of methanol (25%, 50%, 75%, and 100%) in PBT buffer (PBS containing 0.1% of Tween-20) until the moment of performing the procedure. Whole-mount in situ hybridization was performed following standard procedures. Probe against chicken vinculin was created by Dr. Elisa Martí laboratory by cloning though RT-PCR a sequence of 749 nucleotides corre-sponding to the 5′ region of ggVCL mRNA into pBluescript II vector: 5′-TTGCCGGGGGAAGGCGAGAGGGTACCGAGTTCGAGT CCCGCCGGCCGGGAACTGCCGGTCCGCACCGCCGCAGCCCGC TGAGGGGTGAGGAGTTCGGGGCGGCGTCGTTTCCTTATTTCT TTCTTTTTCTCTCTCGGCGGAGGCTGCGGCTTTCTCCGTGGG GCGCCAACCCGCAGGGACAGAGTTTTCCAGGAGTTTGTCGGA CTTCGGGCTCCGGCCCCCCTGCCCCGCTGCCGCCATGCCCG TCTTCACACGCGCACCATCGAGAGCATCTTGGAGCCCGTGGC TCAGCAGATCTCCCACCTGGTCATCATGCACGAGGAGGGGGA GGTAGACGGCAAGGCCATCCCGGACCTCACCGCCCCCGTGTC GGCCGTGCAGGCCGCTGTCAGCAACCTGGTGCGGGTGGAAAA GAAACTGTGCAGACAACAGAAGACCAGATCTTGAAAAGGGAT ATGCCACCAGCATTCATCAAAGTAGAGAATGCCTGCACCAAG CTCGTTCGAGCAGCCCAGATGCTGCAAGCAGATCCTATTCAG TACCAGCTCGTGACTACCTAATTGATGGATCAAGAGGCATCC TTTCTGGAACATCAGACTTACTTCTGACATTTGATGAAGCGG AGGTAGGAATCCTAACACGAGTGACTTAGGTCCGTAAATCAT CCGTGTCTGCAAAGGAATATTGGATATCTGACTGTGGCAGAA GTAGTAGAGACTATGGAGGATTTGGTGACATATACCAAAGA-3′.

Hybridized embryos were post-fixed in 4% PFA, embedded in 5% agarose–10% sucrose blocks and then sectioned with a Vi-bratome (VT1000S; Leica) at 60 μm thickness. Sections were photographed with Olympus DP72 digital color camera attached to a Nikon E600 microscope. The data show a representative image obtained from three embryos.

### Quantifications
#### Actin cytoskeleton disorganization
NT transverse sections that presented a discontinuity in the apical phalloidin staining of at least 10% of the electroporated area were considered as sections with disorganized cytoskeleton. The analysis was performed on 6–76 sections from at least two inde-pendent experiments using three or four embryos per condition.

#### Apicobasal actin distribution
To analyze the actin distribution along the apicobasal axis, pixel intensity measurements of the region of interest were per-formed with the tool plot profile from ImageJ. The apicobasal distance in the images with a similar level of transfection was normalized to 100 pixels. The line plots represent the relative pixel intensity profiles of phalloidin in each normalized image. The analysis was performed on seven sections from two inde-pendent experiments using at least three embryos per condition.

#### Distribution of GFP⁺ nuclei along the apicobasal axis
The apicobasal distance in the images with a similar level of transfection was normalized to 100 pixels. The distance between each nucleus and the apical pole was measured with ImageJ by drawing a straight line from the apical membrane to the nucleus center. The *n* of each experiment is indicated in the figure legend.

### Centrosome positioning

A straight line spanning from each centrosome (Cep152-GFP signal) to the apical NT edge was manually drawn and the distance was measured using ImageJ. The $n$ of each experiment is indicated in the figure legend.

### Ex-vivo chicken NT slice culture and time-lapse imaging

Embryos were electroporated at HH12 stage with pSUPER-shControl or pSUPER-shVCL (vectors without GFP as a reporter) plus mbGFP and H2B-RFP vectors to visualize the plasma membrane and the nucleus, respectively. After 8–10 h of incubation, GFP transfected areas were dissected in ice-cold L15 medium (L4386; Sigma-Aldrich) and embedded in 3% low-gelling-temperature agarose dissolved in neurobasal medium (Invitrogen) to be later sectioned at 250 µm thickness with a vibratome (VT1000S; Leica). Next, selected sections were placed on a 35-mm glass-bottom Petri dish (Ibidi) and embedded in 0.5% low-gelling-temperature agarose dissolved in a neurobasal medium. Slices were cultured in 2 ml of neurobasal medium supplemented with B-27 (#17504044; Thermo Fisher Scientific) and 2 mM L-glutamine. Blebbistatin dissolved in DMSO was used at 12.5 µM and an equal volume of DMSO was used as the control. For time-lapse imaging, images were acquired every 5 min for 10–14 h using the Zeiss LSM-780 microscope (confocal and multiphoton microscope) with a 40× objective (oil, NA 1.3). The culture was maintained at 37.5°C in a humid atmosphere (95% air, 5% $CO_2$). At 5–7 min intervals, 20–25 Z-axis optical sections spaced at 1.5 µm were imaged with a resolution of 1,024 × 1,024 pixels (pixel size 0.31 × 0.31 µm, with a 4 × 4 binning). Images were processed with the ImageJ/Fiji software. For slice cultures, slices were incubated in 95% air, 5% $CO_2$ atmosphere at 37.5°C for 14 h, and fixed overnight in 4% PFA at 4°C.

### Analysis of nuclear migration kinetics

Cell trajectories were generated by tracking the X,Y (X = apicobasal axis; Y = dorso-ventral axis) position occupied by the GFP+ nuclei at the different time points (every 5 min) using the tool MTrackJ of Image/Fiji software. The apical border was given the value Y = 0 at the beginning of each measurement, the Y position was corrected by the corresponding factor in all images where whole tissue movements occurred. To calculate the mean velocity (mV), we divided the total distance travelled by each cell (tD) by the total elapsed time (T). To calculate the directional velocity (dV), we divided the strait distance between the first and last cell position (sD) by T. To calculate the directionality ratio (dR), we divided sD by tD; this ratio would be 0 if tD were infinite and would 1 if tD were equal to sD (Richardson et al., 2016). The X components of trajectory positions were plotted (Y axis) against time (X axis) to generate the trajectory plots shown in Figs. 4 and 5. In these plots, the position of basal (Fig. 4) and apical (Fig. 5) borders was arbitrarily given the value Y = 0.

### EdU incorporation

For EdU incorporation, 200 µl of EdU solution (1 mM) was added onto the vitelline membrane 4 h before the embryos were removed from the egg at stage HH18 and fixed overnight in 4% PFA. The 60-µm sections obtained with the vibratome were processed for EdU detection according to the instructions of the manufacturer of the Click-iT EdU Alexa Fluor 647 Imaging kit (#C10340; Invitrogen). Once the protocol was finished, the sections were mounted on slides for later observation in a Leica SP5 or a Zeiss LSM-780 confocal microscope.

### Flow cytometry analysis

HEK293 cells, HeLa cells, or HH-12 chicken embryos were electroporated with pSHIN-shControl or pSHIN-shVCL plasmids for 24 h. A single-cell suspension was obtained by treating each 10-cm cell dish or 10 embryos (HH18) with 1 ml of 1X Trypsin-EDTA solution (Sigma-Aldrich) containing 100 U of DNAseI for 5–10 min at 37°C. Digestion was stopped by adding one volume of culture media containing 10% of FBS. Individual cells were collected by centrifugation at 300 $g$ for 5 min. Cells were resuspended in 300 µl of PBS and fixed for 15 min adding 1 ml of 4% PFA. PFA was blocked by adding 1 vol of PBS containing 10% of FBS and 0.1% Triton X-100. Cells were centrifuged at 300 $g$ for 5 min and then resuspended in PBS-BSA containing Hoechst and RNase A. Hoechst and GFP fluorescence were determined by FACSAria Fusion cytometer (BD Biosciences) and the data were analyzed with FlowJo software (Tree Star) and Multicycle software (Phoenix Flow Systems; cell cycle profile analysis).

### Statistical analyses

GraphPad Prism 6 software was used for statistical analysis. In all experiments, a normality study of the data was carried out through the D'agostino & Pearson test. In most analyses, data presented a normal distribution, thus the differences were analyzed using the Student's $t$ test when two groups were compared and by one-way ANOVA followed by a Tukey test when three groups or more were compared. Data from measurements of nuclei distribution along the apicobasal axis and the analysis of nuclear migration kinetics did not follow a normal distribution, in this case, the data were analyzed using the U Mann–Whitney test when two groups were compared and by the Kruskal–Wallis test when three or more groups were compared. Quantitative data with normal distribution are expressed as the mean ± standard error of the mean (SEM). Quantitative data with a non-normal distribution are expressed with the median. In all cases, a degree of significance was established with P value <0.05: (*), P value <0.01 (**), P value <0.01 (***), and P < 0.0001 (****). A confidence limit >95% was established.

### Online supplemental material

Fig. S1 shows an assessment of the efficiency of the sh-RNAs used to knockdown the studied ABPs. Fig. S2 shows a demonstration that vinculin requires interaction with actin to stabilize the apical-acting complexes. Fig. S3 shows a demonstration that adherent junctions are not altered after 24 h of vinculin suppression. Fig. S4 shows a demonstration that vinculin is not required for cell cycle progression in non-polarized cells. Table S1 shows the kinetic parameters of INM calculated using the cell trajectories shown in Figs. 4, 5, and 6. Video 1 shows a 360° view of "En face" images of chicken NTs transfected with Cep152-GFP, β-Catenin·Flag, and Arl13b-RFP. Video 2 shows a complete time series of AB[INM] in chicken NTs treated with 1DMSO and

transfected with shControl. Video 3 shows a complete time series of AB<sup>INM</sup> in chicken NTs treated with 12.5 µM of blebbistatin and transfected with shControl. Video 4 shows a complete time series of AB<sup>INM</sup> in chicken NTs treated with DMSO and transfected with shVCL. Video 5 shows a complete time series of BA<sup>INM</sup> in chicken NTs treated with DMSO and transfected with shControl. Video 6 shows a complete time series of BA<sup>INM</sup> in chicken NTs treated with 12.5 µM of blebbistatin and transfected with shControl. Video 7 shows a complete time series of BA<sup>INM</sup> in chicken NTs treated with DMSO and transfected with shVCL. Video 8 shows a 360° view of an internalizing centriole labeled with vinculin-mCherry.

### Data availability
The data reported in this article are available in the published article and its online supplemental material. The plasmids generated in this study are available from the corresponding author upon request.

## Acknowledgments
The authors are indebted to E. Rebollo for her invaluable technical assistance at the AFMU Facility (IBMB). A. Herrera was supported by a Juan de la Cierva fellowship, MINECO #FJCI-2015-26175. A. Ochoa was supported by a MEC pre-doctoral fellowship, MINECO #BES-2015-072035. The work in S. Pons's laboratory was supported by MINECO grant BFU2017-83562-P and MICIN grant PID2020-116806GB-I00. Open Access funding provided by CSIC.

Author contributions: A. Ochoa conceived and performed most experiments, analyzed the data, and discussed the results. A. Herrera conceived and performed experiments, supervised the work, discussed the results, and wrote the manuscript. A. Menendez performed experiments and provided technical support for all the experiments. M. Estefanell performed experiments. C. Ramos performed experiments S. Pons conceived experiments, discussed the results, and wrote the manuscript.

Disclosures: The authors declare no competing interests exist.

Submitted: 28 June 2021

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

# Supplemental material

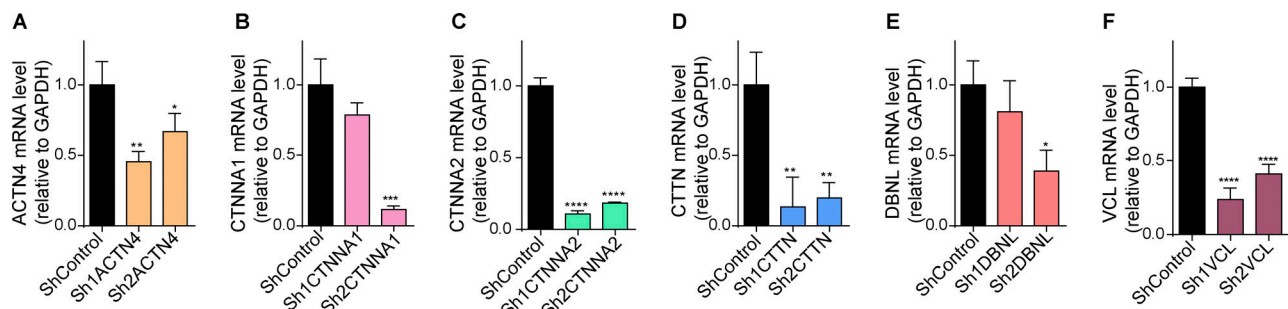

Figure S1. **Efficiency of the sh-RNAs used to knock-down the studied ABPs. (A–F)** The Sh-RNAs targeting each of the six ABPs studied were cloned into pSHIN vector. Two different Sh-RNAs were designed for each gene. The knockdown efficiency was tested by RT-qPCR in chicken embryonary fibroblasts (CEFs). Mean ± SEM. One-way ANOVA plus Dunnett's multiple comparisons test, $n$ = 3 (3 RT-qPCR reactions using three independent CEFs transfections) * = $P < 0.005$, ** = $P < 0.01$, *** = $P < 0.001$, **** = $P < 0.0001$.

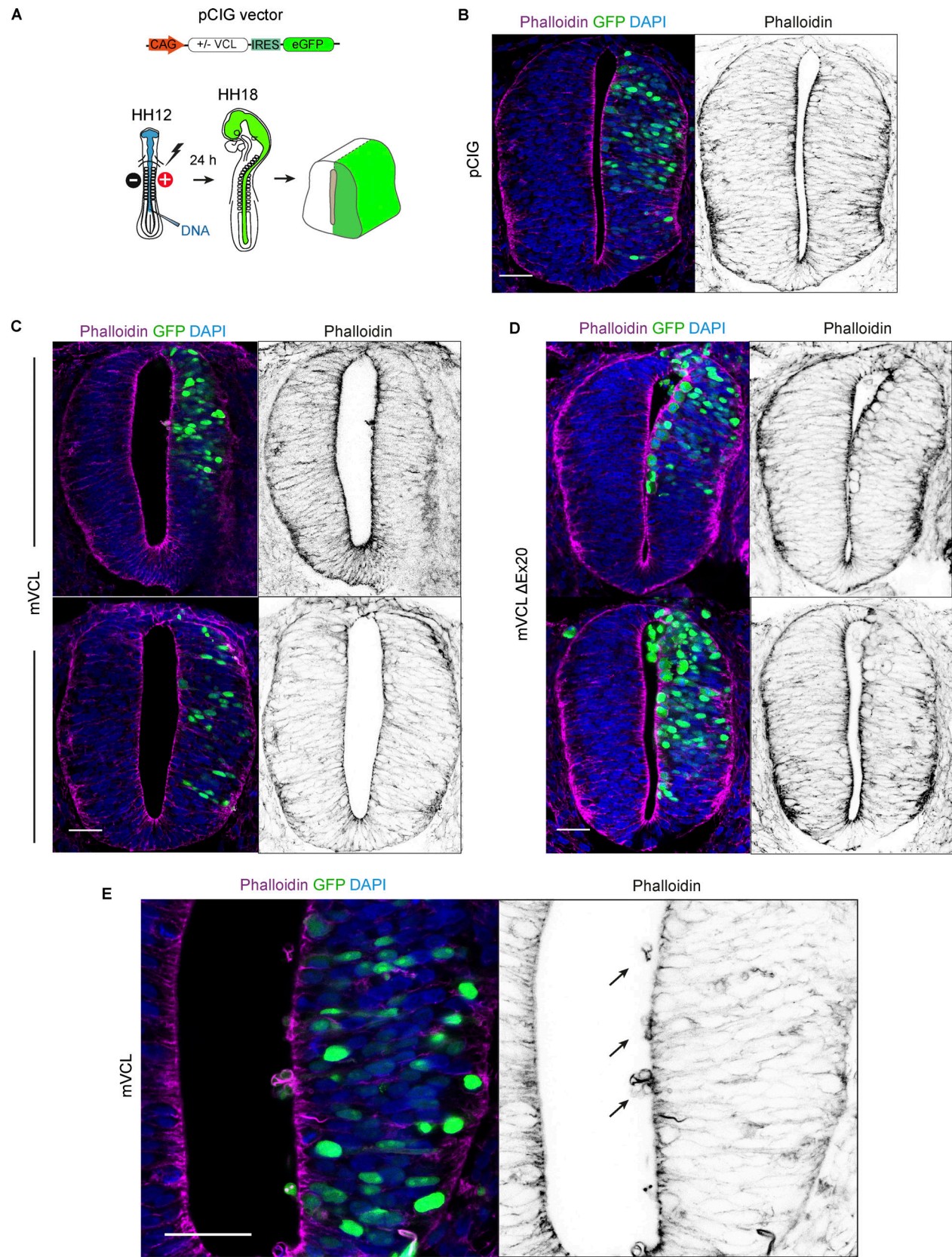

Figure S2. **Vinculin requires the interaction with actin to stabilize the apical acting complexes. (A)** Scheme of the DNA vector and the procedures followed in this figure. **(B–D)** Images of transverse sections of chicken NTs transfected at stage HH12 for 24 h with empty vector, mVCL, or mVCL-ΔEx20 and stained with phalloidin (magenta or gray), nuclear GFP indicates transfection. C and D show two different slices for each construct. Scale bar = 50 µm. **(E)** Higher magnification caption of a slice transfected with mVCL. Actin rings protruding into the ventricle are indicated with arrows.

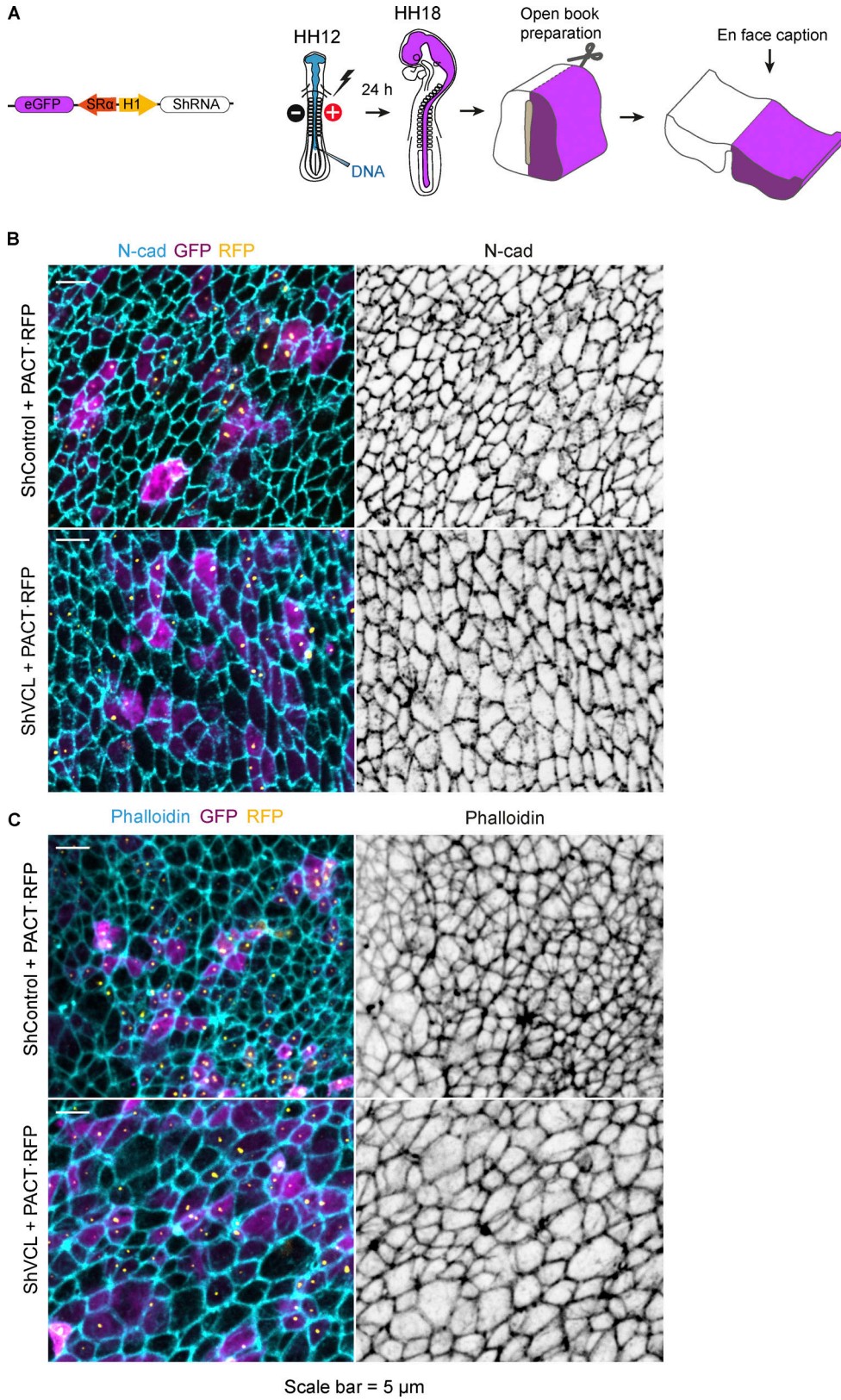

Figure S3. **Adherent junctions are not altered after 24 h of vinculin suppression. (A)** Scheme of the DNA vector used and the procedures followed in this figure. **(B and C)** Open-book captions of chicken NTs transfected at stage HH12 for 24 h with ShControl or ShVCL and PACT·RFP (labels centrioles, yellow) and stained with N-cadherin or phalloidin (cyan or gray). Cytoplasmic GFP (magenta) indicates transfection. Scale bar = 5 µm.

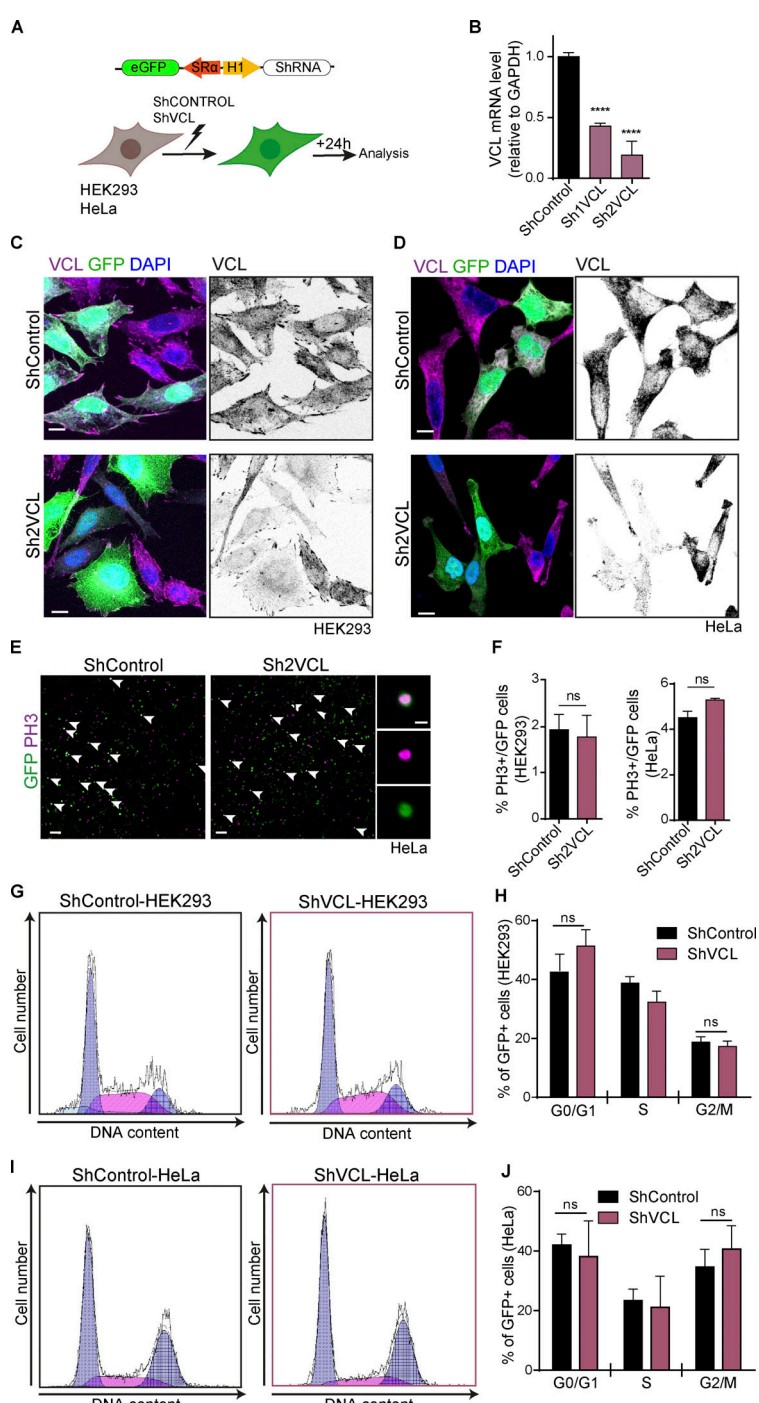

Figure S4. **Vinculin is not required for cell cycle progression in non-polarized cells. (A)** Scheme of the DNA vector used and the procedures followed in this figure. **(B)** Bar graph showing the mRNA level of vinculin measured by RT-qPCR 24 hpe of two different shRNAs targeting human vinculin into HEK-293 cultures. Mean ± SEM. One-way ANOVA plus Dunnett's multiple comparisons test, n = 3 (3 RT-qPCR reactions using three independent HEK-293 transfections). **(C and D)** HEK-293 (C) and HeLa (D) cells transfected with shControl or sh2VCL, stained with anti-vinculin (magenta or gray scale), GFP (green) indicates transfection, and DAPI (blue) shows the nucleus. Scale bar = 10 μm. **(E)** Images of HeLa cultures transfected with shControl or sh2VCL, stained with anti PH3 (magenta); GFP (green) shows transfection. Arrowheads indicate transfected mitotic cells (PH3+/GFP+). Scale bar = 100 μm. The right panels show a magnification of an example of a transfected mitotic cell (PH3+/GFP+), the top panel shows PH3 and GFP channels together and the two bottom ones show them separately. Scale bar = 10 μm. **(F)** Bar graphs showing the percentage of mitotic (PH3+) among the transfected population (GFP+) in HEK-293 (left graph) or HeLa (right graph) cultures transfected with shControl or sh2VCL. Mean ± SEM. t test, n = 1,438 cells for HEK and n = 12,650 cells for HeLa coming from three independent transfections. **(G)** Flow cytometry analysis profiles showing DNA content (Hoechst fluorescence intensity) and cell number of GFP+ population of HEK293 cultures transfected with shControl or shVCL. A minimum of 5,000 cells were analyzed for each condition. **(H)** Bar graph showing the percentage of cells in each cell cycle phase from the profiles shown in Fig. 8 G. Mean ± SEM. t test, n = 3 independent transfections **(I)** Same experiment as in Fig. 8 G but performed in HeLa cells. **(J)** Bar graph showing the percentage of cells in each cell cycle phase from the profiles shown in Fig. 8 I. Mean ± SEM. t test, n = 3 independent transfections. ns = non-significant. **** = P < 0.0001.

Video 1.  **360° view of "en face" images of chicken NTs transfected with Cep152-GFP, β-Catenin·Flag, and Arl13b-RFP.** Chicken NTs transfected at stage HH12 for 24 h with Cep152-GFP (green), β-Catenin·Flag (magenta), and Arl13b-RFP (cyan). Nuclei were stained with DAPI (blue). The four channels are shown first together and then separately. 1° angle progression at 30 fps. The corresponding 2D images are shown in Fig. 2 G.

Video 2.  **Complete time-series of AB^INM in chicken NTs treated with DMSO and transfected with shControl.** Chick NTs electroporated at HH12 stage with shControl plus mbGFP (green) and H2B-RFP (magenta) vectors to visualize the plasma membrane and the nucleus, respectively, and treated with DMSO. After 8–10 h of incubation, GFP transfected areas were dissected and sectioned at 250-μm thickness. Confocal images of the slices were acquired every 5 min for 10–14 h using the Zeiss LSM-780 microscope with a 40× objective. The white line shows the trajectory of a recorded cell. Reproduced at 30 fps. Selected video stills are shown in Fig. 4 A.

Video 3.  **Complete time-series of AB^INM in chicken NTs treated with 12.5 μM of blebbistatin and transfected with shControl.** Chick NTs electroporated at HH12 stage with shControl plus mbGFP (green) and H2B-RFP (magenta) vectors to visualize the plasma membrane and the nucleus, respectively, and treated with 12.5 μM of blebbistatin. After 8–10 h of incubation, GFP transfected areas were dissected and sectioned at 250-μm thickness. Confocal images of the slices were acquired every 5 min for 10–14 h using the Zeiss LSM-780 microscope with a 40× objective. The white line shows the trajectory of a recorded cell. Reproduced at 30 fps. Selected video stills are shown in Fig. 4 B.

Video 4.  **Complete time-series of AB^INM in chicken NTs treated with DMSO and transfected with shVCL.** Chick NTs electroporated at HH12 stage with shVCL plus mbGFP (green) and H2B-RFP (magenta) vectors to visualize the plasma membrane and the nucleus, respectively, and treated with DMSO. After 8–10 h of incubation, GFP transfected areas were dissected and sectioned at 250 μm thickness. Confocal images of the slices were acquired every 5 min for 10–14 h using the Zeiss LSM-780 microscope with a 40× objective. The blue line shows the trajectory of a recorded cell. Reproduced at 30 fps. Selected video stills are shown in Fig. 4 C.

Video 5.  **Complete time-series of BA^INM in chicken NTs treated with DMSO and transfected with shControl.** Chick NTs electroporated at HH12 stage with shControl plus mbGFP (green) and H2B-RFP (magenta) vectors to visualize the plasma membrane and the nucleus, respectively, and treated with DMSO. After 8–10 h of incubation, GFP transfected areas were dissected and sectioned at 250 μm thickness. Confocal images of the slices were acquired every 5 min for 10–14 h using the Zeiss LSM-780 microscope with a 40× objective. The white line shows the trajectory of a recorded cell. Reproduced at 30 fps. Selected video stills are shown in Fig. 5 A.

Video 6.  **Complete time-series of BA^INM in chicken NTs treated with 12.5 μM of blebbistatin and transfected with shControl.** Chick NTs electroporated at HH12 stage with shControl plus mbGFP (green) and H2B-RFP (magenta) vectors to visualize the plasma membrane and the nucleus, respectively, and treated with 12.5 μM of blebbistatin. After 8–10 h of incubation, GFP transfected areas were dissected and sectioned at 250 μm thickness. Confocal images of the slices were acquired every 5 min for 10–14 h using the Zeiss LSM-780 microscope with a 40× objective. The white line shows the trajectory of a recorded cell. Reproduced at 30 fps. Selected video stills are shown in Fig. 4 B.

Video 7.  **Complete time-series of BA^INM in chicken NTs treated with DMSO and transfected with shVCL.** Chick NTs electroporated at HH12 stage with shVCL plus mbGFP (green) and H2B-RFP (magenta) vectors to visualize the plasma membrane and the nucleus, respectively, and treated with DMSO. After 8–10 h of incubation, GFP transfected areas were dissected and sectioned at 250 μm thickness. Confocal images of the slices were acquired every 5 min for 10–14 h using the Zeiss LSM-780 microscope with a 40× objective. The blue line shows the trajectory of a recorded cell. Reproduced at 30 fps. Selected video stills are shown in Fig. 4 C.

Video 8.  **360° view of an internalizing centriole labeled with vinculin-mCherry**. The transverse section of a chicken NT transfected at stage HH12 for 24 h with geminin·AG (green), VCL·mCherry (magenta), and Cep152·GFP (green) and stained with N-cadherin (cyan). The three channels are shown first together and then separately. The image shows an internalizing centrosome at the apical portion of a NSC. 1° angle progression at 30 fps. The corresponding 2D image is shown in the lower panel of Fig. 8 C.

**Provided online is Table S1. Table S1 shows kinetic parameters of INM calculated using the cell trajectories shown in Figs. 4, 5, and 6.**

