## [Peer Review File · The Journal of Cell Biology]

Vinculin is required for interkinetic nuclear migration (INM) and cell cycle progression

ANDREA OCHOA, Antonio Herrera, ANGHARA MENENDEZ, Maria Estefanell, Carlota Ramos, and Sebastián Pons

Corresponding Author(s): Sebastián Pons, Molecular Biology Institute of Barcelona and Antonio Herrera, Molecular Biology Institute of Barcelona

Review Timeline:

Submission Date:	2021-06-28
Editorial Decision:	2021-08-26
Revision Received:	2023-08-08
Editorial Decision:	2023-09-26
Revision Received:	2023-10-06

Monitoring Editor: Marianne Bronner

Scientific Editor: Dan Simon

Transaction Report:

DOI: <https://doi.org/10.1083/jcb.202106169>

August 26, 2021

Re: JCB manuscript #202106169

Dr. Sebastián Pons
Molecular Biology Institute of Barcelona
Department of Cell Biology
C/ Baldiri Reixac, 10-12
Edificio Cluster, Parc Científic de Barcelona
Barcelona, Barcelona 08028
Spain

Dear Dr. Pons,

Thank you for submitting your manuscript entitled "Vinculin is required for interkinetic nuclear migration (INM) and cell cycle progression." The manuscript has been evaluated by expert reviewers, whose reports are appended below. You will see that the reviewers feel that the work is well done and find the link between vinculin and interkinetic nuclear migration intriguing. However, they also note that the study does not provide significant insights into the molecular mechanisms by which vinculin mediates both interkinetic nuclear migration and centrosome internalization. For this reason, regrettably, we are unable to accept the present version of your manuscript.

Although your manuscript is intriguing, we feel that the points raised by the reviewers are more substantial than can be addressed in a typical revision period. If you feel you are able to provide more mechanistic insight as suggested by the reviewers, we would be open to resubmission of a significantly revised version of the manuscript that fully addresses the reviewers' concerns and is subject to further peer-review. If you would like to resubmit this work to JCB, please contact the journal office to discuss an appeal of this decision or you may submit an appeal directly through our manuscript submission system. Please note that priority and novelty would be reassessed at resubmission.

If you wish to expedite publication of the current data, it may be best to pursue publication at another journal. Our journal office can transfer your reviewer comments to another journal upon request.

Regardless of how you choose to proceed, we hope that the comments below will prove constructive as your work progresses. We would be happy to discuss the reviewer comments further once you've had a chance to consider the points raised in this letter. You can contact the journal office with any questions, cellbio@rockefeller.edu or call (212) 327-8588.

Thank you for thinking of JCB as an appropriate place to publish your work.

Sincerely,

Marianne Bronner, PhD
Monitoring Editor
Journal of Cell Biology

Dan Simon, PhD
Scientific Editor
Journal of Cell Biology

Reviewer #1 (Comments to the Authors (Required)):

This work studies the role of the adhesion protein vinculin in the interkinetic nuclear migration (INM) of chicken neural tube. They show that inhibition of vinculin affects apical actin localization, centrosome distribution, cell cycle progression and finally INM.

This is an interesting work that links vinculin with several processes that take place during INM. The manuscript is well written, and the data well presented; however, there is no clear mechanistic explanation that links in a causal manner all these events affected by vinculin depletion.

Specific comments:

Fig 1: Certainly knock down of vinculin produces the strongest phenotype in the neural tube; however loss of α NCatenin,

Cortactin and Debrin-like generate serious malformations in the electroporated neural tube. In addition, the authors emphasise the observation that in the vinculin LOF experiment "the characteristic actin accumulation forming mainly the apical actin belt was entirely erased"; however, a similar erasing of actin in the apical actin belt was observed when Debrin-like was depleted. These phenotypes seem ignored in the rest of the manuscript.

Fig 2: In order to study the localization of vinculin in the neural tube, the authors express a mutated form of vinculin-GFP, with reduced self-interaction. They need to expand more about this mutant. What is the evidence that does not affect localization into FA? Previous publications? In addition, higher magnifications are required to identify where (AJs?) vinculin is located.

Based on the data provided in Figs 4-6, the authors conclude "these results demonstrate that in chicken NT epithelium, INM is driven by acto-myosin contraction acting during BA INM and that vinculin is required for this process". This is an overinterpretation of their results. Firstly, although blebbistatin is an inhibitor of myosin, it is not a specific inhibitor of actomyosin contraction; for example, actomyosin is required for cell-cell adhesion, which is known to be inhibited by blebbistatin. Secondly, vinculin is known to play many cellular functions, being its involvement in actomyosin contraction only one of them. The authors need to provide more direct evidence that support their claims or soften this conclusion.

It is not clear how the authors can distinguish the function of vinculin during INM between cell-cell adhesion (AJ) or cell-matrix adhesion (FA). As the vinculin-A50I mutant affect the interaction with talin (FA) and b-catenin (FA and AJ); in addition, vinculin localization does not allow to distinguish between these two adhesion structures, as the apical localization of vinculin could be consistent with a function in FA or AJ.

The authors show that the basal-to-apical nuclear migration was affected in vinculin depleted neural tube, whereas the opposite basal-to-apical remains largely normal. At the same time the authors describe the loss of apical actin as one of the major effects of vinculin inhibition. How are these two observations concealed? What is the mechanism by which apical actin is involved in basal-to-apical movement? High resolution images of actin could help to solve this issue.

Vinculin inhibition impaired INM and centrosome positioning. Is there any causal link between these two processes? Which is the order of events? Is actin disruption affecting centrosome position? Or is tubulin disruption impairing actin apical accumulation?

Fig 8 is almost a control to show the specificity of vinculin function in neural tube morphogenesis and it distracts from the focus of the paper. It could be moves as a supplementary figure.

Reviewer #2 (Comments to the Authors (Required)):

The paper highlights the role of vinculin in controlling a fundamental neurogenetic event during chick embryo development, as well as how the function of the protein interconnects with the intricate dynamics of the actin cytoskeleton, intracellular contractile forces and key components of the neuroepithelial junctional complexes.

More specically, the described discoveries make an important contribution to the understanding of the regulation of the interkinetic nuclear migration that neural (epithelial) stem cells ensue to shape the CNS.

The importance of vinculin in this neurogenetic process could have been inferred from previously published data, alongside our knowledge about the role played by the molecule in the modulation of the cytoskeletal and contractile mechanics dictated by cell-cell and cell-matrix interactions. However, a clear involvement of vinculin in the cell cycle-associated NSC nuclear displacement has not previously been reported in vivo. The authors are documenting this involvement in a clear and rather convincing way, thereby warranting publication of the manuscript after some minor modifications.

My first suggestion to the authors is to reorganize some parts of the manuscript to render it less discursive in the Result sections and better structured for a more fluid reading and better grasping of the experimental approaches. A discussion on "vinculin" would be appropriate in the Introduction (and not in a "Results" paragraph), whereas the redundant "second Abstract" at the end of the "Introduction" can be omitted.

Several of the paragraphs of the Results need to be concised and streamlined. An effort should also be made to better structure the Discussion.

There are further some experimental issues that need to be better elaborated.

1. The relative levels of knockdown of vinculin obtained in vivo after in ovo electroporation of the shRNA probe should be documented, preferably at different time points, to give a better idea of how much of the vinculin pool engages in the process (as also related to the assumption made in the second paragraph of the Results; Fig. 3A).

This may also give hints about the 24 hrs-delay in the modulation of the actin cytoskeleton after vinculin depletion, while nuclear displacement was affected.

2. Given the massive effect observed after vinculin knockdown, it would be useful to incorporate a rescuing experiment, to further substantiate the specificity of the *in vivo* knockdown. Such type of experiment is documented in the context of cell-cycle progression (Fig. 9), but not in the context of the cell arrangement within the NT.
3. The schematic drawings in Fig. 4F, reproducing the cell-cycle dependent nuclear displacement monitored by video time-lapse after vinculin knockdown, illustrates a behaviour largely analogous to that seen in the control treated NTs. Yet, the plots reporting the distance measurements show differences in the nuclear patterns. The authors should elaborate on this apparent discrepancy.
4. The vinculin-myosin II interconnection is intriguing and calls upon a verification through a dominant-negative approach entailing ectopic overexpression of vinculin mutated for its actin-binding properties.
5. The control/reference vinculin knockdown experiments in human cell lines might be appropriate, but unfortunately as these experiments were performed are poorly informative. First, the choice of cell line was not optimal because of the incomplete epithelial phenotype of the chosen cells. A cell line with a more accentuated non-neoplastic epithelial phenotype (there are several such cells to access) would be preferred.

Moreover, to make conclusions about the essentiality of cell polarity for the involvement of vinculin in cell-cycle progression, 3D geometries have to be considered, i.e. it is not sufficient to just analyze sparse cells in a monolayer configuration. For instance, cadherin-mediated cell-cell contacts may be more pivotal than a putative apical-to-basal polarity, as the findings reported by the authors in Fig. 9 would underscore.

Reviewer #3 (Comments to the Authors (Required)):

This paper aims to understand how several well-known cellular processes, including the formation of adherens junctions at the apical side of neural tube, interkinetic nuclear migration, cell cycle progression, centrosome internalization, are coordinated to control morphogenesis of neural stem cells in the embryonic neural tube. Using embryology, immunofluorescence, live imaging, and quantitative analysis, the authors show vinculin is required for all these cellular processes. While this work clearly shows the essential role of vinculin in neural tube development and suggests it is potentially a new molecular regulator of all these cellular processes, it does not dig deep into the regulatory mechanisms of any of these processes and it does not show whether and how vinculin functions as a link.

All the studies of BAINM, cell cycle progression, and centrosome internalization on the shVCL expressing embryos were performed in a stage (within 24h after transfection) before the disruption of actin and adherens junction structure (after 36h after transfection). I was curious about the motivation of this experimental design. Do the authors want to argue these processes are critical for the maintenance of adherens junctions? If so, the paper does not provide data to support a casual relationship.

The paper needs to have a schematic diagram to clearly summarize the relationship of all the processes and vinculin at a single cell level.

Below are my comments on individual figures:

Fig 1.

1J, J': In J, there are two very strong F-actin stripes across the middle of the transfected neural tube and it seems there are more cells also. Consistently, J' shows a redistribution of F-actin on the transfected side. I am curious it is a new function of vinculin, or it is a secondary effect from other defective processes.

Fig 2.

2B: It seems there are more vinculin proteins on the apical side, but they are very sparsely distributed. Is that due to the dynamic regulation of the polarized distribution of vinculin or the transfection efficiency? If adherens junction is the anchor for cell attachment and vinculin is an important component of it, the cells highly expressing VCL-mCherry on the apical side should have more active BAINM. This can be tested by live imaging on the cells expressing VCL-mCherry and H2B-GFP.

2C: The optical resolution is not sufficient to argue a colocalization between vinculin and Ncad. To clearly show vinculin is the component of adherens junction, it is important to show one high resolution image (using 63x objective lens with high NA) of VCL-mCherry and N-Cad staining at single cell level.

Fig 4-6.

In Materials and Methods, it is stated the live imaging was performed 8-10 hours after transfection of shVCL. This should be clearly stated in Results so the readers know how to interpret the data.

Need to add scale bars to the images in 4A-C, 5A-C and all videos.

The authors show velocity and directionality changes in myosin and vinculin knock down cells, but I also observed another interesting phenomenon. In video 5 (BAINM in Bleb treated cells), the authors label one nuclei that was moving around in the middle of the tissue, but below this nuclei there was another one which also initially moved around in the middle of the tissue and then moved smoothly to the apical side from 4:05-4:35. The moving behavior of the second nuclei in this video is similar the shVCL transfected cell in video 6 (BAINM in shVCL treated cells). In contrast, the movement of the normal cells is smoother in video 4. This is consistent with the trajectory analysis. If you compare 5F with 5G and H, indeed the normal cells have straighter trajectories, while the cells treated with Bleb or expressing shVCL show zigzag trajectories and become a little bit straighter over time. Is it possible that this phenotype is more directly relevant to the absence of vinculin while abnormal cell velocity and directionality are the secondary effects? Or could it be just due to a compression of neighboring cells? Why do the nuclei move more straighter later on? Are the zigzag trajectories caused by the defective internalization of centrosomes to pull the nuclei through tubulin or does it due to the loose adherens junction? Is the interaction between vinculin and myosin direct or indirect? Whether and how this migration behavior affects adherens junctions? In my view, addressing any of these questions using the imaging system the authors established will deepen our understandings of the cell biology in vivo.

Fig 7.

I am a little bit disappointed that the authors did not use vinculin as a key reagent to further test the relationships between cell cycle progression, centrosome internalization, and INM.

7L-K: I have another hypothesis: one centrosome internalization may establish tubulin (one end binds to the centrosome connecting to the nuclei; the other end binds to the centrosome connecting to the adherens junction) to pull the nucleus to the apical side. Is it possible that in the vinculin knock down cells the failure of centrosome internalization blocks the formation of this tubulin structure; as a result, the nuclei show a defective and zigzag BAINM?

Fig 9.

Does disrupting the normal structure of adherens junctions in Ncad mutants produces similar phenotype?

The experiments in Fig3-7 were performed in the embryos where the adherens junctions still have normal structure (apical side of the transfected neural tube looks normal). The phenotypes in INM and cell cycle progression observed in those experiments suggest some other functions of vinculin rather than its binding to adherens junctions, are required for these cellular processes. Or is it possible that in those experiments the adherens junction structure looks normal but is loose? If so, the authors need to show that.

Reviewer #1 (Comments to the Authors (Required)):

R

This work studies the role of the adhesion protein vinculin in the interkinetic nuclear migration (INM) of chicken neural tube. They show that inhibition of vinculin affects apical actin localization, centrosome distribution, cell cycle progression and finally INM.

This is an interesting work that links vinculin with several processes that take place during INM. The manuscript is well written, and the data well presented; however, there is no clear mechanistic explanation that links in a causal manner all these events affected by vinculin depletion.

Specific comments:

Fig 1: Certainly knock down of vinculin produces the strongest phenotype in the neural tube; however loss of α N-catenin, Cortactin and Debrin-like generate serious malformations in the electroporated neural tube. In addition, the authors emphasise the observation that in the vinculin LOF experiment "the characteristic actin accumulation forming mainly the apical actin belt was entirely erased"; however, a similar erasing of actin in the apical actin belt was observed when Debrin-like was depleted. These phenotypes seem ignored in the rest of the manuscript.

A

Absolutely agree with the reviewer. In this work, our focus was on describing the activity of VCL on various aspects of cell physiology. We are fully aware that many, if not all, of the other actin-binding proteins (ABPs) identified in the screening mentioned in Herrera et al. 2021 deserved more attention. However, the task was too ambitious to be tackled in a single work. In particular, the contribution of Debrin-like, as noted by the reviewer, was remarkable. In fact, it was thoroughly studied and was published in Herrera et al. 2021.

R

Fig 2: In order to study the localization of vinculin in the neural tube, the authors express a mutated form of vinculin-GFP, with reduced self-interaction. They need to expand more about this mutant. What is the evidence that does not affect localization into FA? Previous publications? In addition, higher magnifications are required to identify where (AJs?) vinculin is located.

A

We fully agree with the reviewer's assessment that the construct used was not ideal for inferring vinculin location in the neural epithelium or for studying its precise localization in detail. Higher magnifications were indeed necessary. However, despite these limitations, we made the decision to keep the low magnification images of the VCL-T12 construct in order to provide a general overview of vinculin localization. We have also included a reference describing this construct. Furthermore, in response to the reviewer's suggestions, we have now incorporated higher magnification images in our study. These images are presented in Figure 2 F,G and, more importantly, in Figure 8C. In these figures, we describe the localization of wild-type vinculin in relation to N-cadherin (adherens junctions or AJs) as well as centrosome and primary cilium markers. These higher magnification images provide a more detailed understanding of vinculin localization in the context of specific cellular structures.

R

Based on the data provided in Figs 4-6, the authors conclude "these results demonstrate that in chicken NT epithelium, INM is driven by acto-myosin contraction acting during BA INM and that vinculin is required for this process". This is an overinterpretation of their results. Firstly, although blebbistatin is an inhibitor of myosin, it is not a specific inhibitor of actomyosin contraction; for example, actomyosin is required for cell-cell adhesion, which is known to be inhibited by blebbistatin. Secondly, vinculin is known to play many cellular functions, being its involvement in actomyosin contraction only one of them. The authors need to provide more direct evidence that support their claims or soften this conclusion.

A

We agree with the reviewer's point that we have not provided evidence demonstrating that vinculin's role is dependent on its involvement in actomyosin contraction. As a result, we have modified the sentence to state that "INM is driven by acto-myosin acting during BA INM, and vinculin is also required for this process." However, in Figure 9 E-S, we demonstrate that vinculin A50I fails to rescue the shVCL, thereby indicating that vinculin's sole cellular function involved in BA INM is its ability to bind to components of cell adhesion complexes.

R

It is not clear how the authors can distinguish the function of vinculin during INM between cell-cell adhesion (AJ) or cell-matrix adhesion (FA). As the vinculin-A50I mutant affect the interaction with talin (FA) and b-catenin (FA and AJ); in addition, vinculin localization does not allow to distinguish between these two adhesion structures, as the apical localization of vinculin could be consistent with a function in FA or AJ.

A

In the neuroepithelium, adherens junctions and focal adhesions are restricted to the apical and basal poles, respectively. In adherens junctions, vinculin associates with β -catenin, while in focal adhesions, it associates with talin. Although it is true that vinculin A50I demonstrates reduced binding to both β -catenin and talin, all the new Vinculin-mCherry images shown in Figure 8C and 9 A-C clearly demonstrate that, at least in the neuroepithelium, vinculin has a preference for the apical pole compared to the basal pole. Furthermore, in the VCL suppression experiments, it is evident that the apical pole of the cells undergoes disruption while the basal pole remains intact. However, we must admit that with the tools and data presented, a direct effect on the focal adhesions of the basal pole cannot be definitively ruled out, which could indirectly interfere with interkinetic nuclear migration or centriole internalization.

R

The authors show that the basal-to-apical nuclear migration was affected in vinculin depleted neural tube, whereas the opposite basal-to-apical remains largely normal. At the same time the authors describe the loss of apical actin as one of the major effects of vinculin inhibition. How are these two observations concealed? What is the mechanism by which apical actin is involved in basal-to-apical movement? High-resolution images of actin could help to solve this issue.

A

In our study, we demonstrate that the suppression of vinculin initially leads to alterations in centriole internalization along with disruptions in baso-apical interkinetic nuclear migration. Subsequently, a generalized deterioration of the apical actin rings is observed, which ultimately affects the overall structure of the neuroepithelium. In the new images presented in Figures 2F and 8C, it is clearly evident that vinculin is localized at the apical

pole with enrichment in the adherens junctions and the centrosomal region. Additionally, this region exhibits a fine mesh of actin that encompasses the entire area, in addition to the actin ring at the apical pole. During VCL suppression, although there may not be initially macroscopic alterations in the actin cytoskeleton, there is a slowdown/inactivation of both centrosome internalization and baso-apical interkinetic nuclear movement. It is therefore logical to assume that the initial manifestations of actin cytoskeleton destabilization after vinculin suppression will primarily affect the functions of this fine actin mesh that encompasses the entire apical pole. Therefore, we propose that while the tubulin pulls the nucleus, as demonstrated by Spear and Erickson (2012), the apical actin ring, through vinculin and myosin, reinforces the anchoring to adherens junctions (AJ) to withstand the force generated by tubulin.

R

Vinculin inhibition impaired INM and centrosome positioning. Is there any causal link between these two processes? Which is the order of events? Is actin disruption affecting centrosome position? Or is tubulin disruption impairing actin apical accumulation?

A

The questions raised by the reviewer above are indeed very interesting. However, addressing them presents technical challenges as experiments with fixed materials are not appropriate for determining the precise sequence of events. Instead, ex vivo experiments and time-lapse recordings are necessary. Nonetheless, obtaining satisfactory results in ex vivo experiments at the required resolution level is currently quite challenging. Nevertheless, both processes share a common requirement of tension generated by actomyosin contraction, which is supported by vinculin.

R

Fig 8 is almost a control to show the specificity of vinculin function in neural tube morphogenesis and it distracts from the focus of the paper. It could be moved as a supplementary figure.

A

Following the reviewer's suggestions, we have relocated the previous Figure 8, which described that vinculin suppression does not affect the cell cycle in non-polarized cultured cells, to the supplementary section. It is now presented as Supplementary Figure 4.

Reviewer #2 (Comments to the Authors (Required)):

R

The paper highlights the role of vinculin in controlling a fundamental neurogenetic event during chick embryo development, as well as how the function of the protein interconnects with the intricate dynamics of the actin cytoskeleton, intracellular contractile forces and key components of the neuroepithelial junctional complexes.

More specifically, the described discoveries make an important contribution to the understanding of the regulation of the interkinetic nuclear migration that neural (epithelial) stem cells ensue to shape the CNS.

The importance of vinculin in this neurogenetic process could have been inferred from previously published data, alongside our knowledge about the role played by the molecule in the modulation of the cytoskeletal and contractile mechanics dictated by cell-cell and

cell-matrix interactions. However, a clear involvement of vinculin in the cell cycle-associated NSC nuclear displacement has not previously been reported in vivo. The authors are documenting this involvement in a clear and rather convincing way, thereby warranting publication of the manuscript after some minor modifications.

My first suggestion to the authors is to reorganize some parts of the manuscript to render it less discursive in the Result sections and better structured for a more fluid reading and better grasping of the experimental approaches. A discussion on "vinculin" would be appropriate in the Introduction (and not in a "Results" paragraph), whereas the redundant "second Abstract" at the end of the "Introduction" can be omitted.

A

Thank you and done!

R

Several of the paragraphs of the Results need to be concised and streamlined.

An effort should also be made to better structure the Discussion.

A

Thank you and done!

R

There are further some experimental issues that need to be better elaborated.

1. The relative levels of knockdown of vinculin obtained in vivo after in ovo electroporation of the shRNA probe should be documented, preferably at different time points, to give a better idea of how much of the vinculin pool engages in the process (as also related to the assumption made in the second paragraph of the Results; Fig. 3A). This may also give hints about the 24 hrs-delay in the modulation of the actin cytoskeleton after vinculin depletion, while nuclear displacement was affected.

A

The reviewer's suggestion is very interesting; unfortunately, none of the approaches to answer this question have yielded satisfactory results. The ideal tool to verify the amount of vinculin remaining at different suppression times, and thus infer how much is needed for each cellular function, would undoubtedly have been an antibody specifically recognizing chicken vinculin. Unfortunately, we have not been able to locate any reliable functioning antibody. Another alternative would have been to study the mRNA levels of vinculin specifically by selecting the population transfected with ShVCL. Unfortunately, changes in mRNA after transfection do not clarify how much vinculin remains in each part of the cell. Nevertheless, we believe that our approach of studying the sequence of events that occur after suppression is the most realistic way to attempt to establish a correlation between the decrease in vinculin and the different cellular processes that are sequentially affected.

R

2. Given the massive effect observed after vinculin knockdown, it would be useful to incorporate a rescuing experiment, to further substantiate the specificity of the in vivo knockdown. Such type of experiment is documented in the context of cell-cycle progression (Fig. 9), but not in the context of the cell arrangement within the NT.

A

In Figure 9 M-S as correctly indicated by the reviewer, we show rescues of vinculin suppression using non-targetable vinculin with ShRNA against chicken vinculin in the

context of the cell cycle. However, using these same constructs we also show in Figure 9 E-K the cell positioning rescues requested by the reviewer.

R

3. The schematic drawings in Fig. 4F, reproducing the cell-cycle dependent nuclear displacement monitored by video time-lapse after vinculin knockdown, illustrates a behaviour largely analogous to that seen in the control treated NTs. Yet, the plots reporting the distance measurements show differences in the nuclear patterns. The authors should elaborate on this apparent discrepancy.

A

Thanks, we have modified the schematic drawing, and we think now it matches better with the data reported in the plots.

R

4. The vinculin-myosin II interconnection is intriguing and calls upon a verification through a dominant-negative approach entailing ectopic overexpression of vinculin mutated for its actin-binding properties.

A

Following the reviewer's instructions, we tested the effects of overexpression of a vinculin mutant with low affinity for actin (VCL- Δ Ex20), comparing its effect with the overexpression of wild-type vinculin or an empty vector. Overexpression of wild-type vinculin led to an accumulation of actin at the apical pole, to the extent that it caused the outward movement of the apical poles. In contrast, expression of the VCL- Δ Ex20 mutant seemed to weaken the apical poles to the point where the cells invaded the ventricle. It is worth noting the similarity between the overexpression of the VCL- Δ Ex20 mutant and the suppression of expression using ShRNAs. In both cases, there is a disruption of the apical poles, although with the VCL- Δ Ex20 mutant, this disruption occurs earlier. This is logical since in the suppression, we must wait for the vinculin reserves to be depleted, whereas with the VCL- Δ Ex20 mutant, the effect on the apical poles is immediate after overexpression. Considering the reviewer's strong interest in these questions, we deemed it appropriate to present all these results in a supplementary figure (Figure S2) and dedicate a paragraph to them in the results section.

R

5. The control/reference vinculin knockdown experiments in human cell lines might be appropriate, but unfortunately as these experiments were performed are poorly informative. First, the choice of cell line was not optimal because of the incomplete epithelial phenotype of the chosen cells. A cell line with a more accentuated non-neoplastic epithelial phenotype (there are several such cells to access) would be preferred.

Moreover, to make conclusions about the essentiality of cell polarity for the involvement of vinculin in cell-cycle progression, 3D geometries have to be considered, i.e. it is not sufficient to just analyze sparse cells in a monolayer configuration. For instance, cadherin-mediated cell-cell contacts may be more pivotal than a putative apical-to-basal polarity, as the findings reported by the authors in Fig. 9 would underscore.

A

We fully agree with the reviewer. We conducted those experiments initially to rule out any toxic effect of vinculin suppression that could potentially affect viability and, consequently, the cell cycle. Additionally, we believed it would provide us with information about vinculin requirements in non-polarized cells growing in 2D culture. We observed that

vinculin suppression does not appear to have any effect on proliferation in the tested cell lines (HEK and HeLa). Unfortunately, as the reviewer rightly points out, these cell lines may be too different from the neural tube to draw more definitive conclusions regarding the necessity of vinculin in polarized epithelia versus non-polarized cells. Therefore, we have decided to be more cautious when interpreting these results. Nonetheless, they still serve as a good control for the lack of toxicity, so we have decided to include them in the study as a supplementary figure (Figure S4).

Reviewer #3 (Comments to the Authors (Required)):

R

This paper aims to understand how several well-known cellular processes, including the formation of adherens junctions at the apical side of neural tube, interkinetic nuclear migration, cell cycle progression, centrosome internalization, are coordinated to control morphogenesis of neural stem cells in the embryonic neural tube. Using embryology, immunofluorescence, live imaging, and quantitative analysis, the authors show vinculin is required for all these cellular processes. While this work clearly shows the essential role of vinculin in neural tube development and suggests it is potentially a new molecular regulator of all these cellular processes, it does not dig deep into the regulatory mechanisms of any of these processes and it does not show whether and how vinculin functions as a link.

All the studies of BAINM, cell cycle progression, and centrosome internalization on the shVCL expressing embryos were performed in a stage (within 24h after transfection) before the disruption of actin and adherens junction structure (after 36h after transfection). I was curious about the motivation of this experimental design. Do the authors want to argue these processes are critical for the maintenance of adherens junctions? If so, the paper does not provide data to support a casual relationship.

A

I would like to emphasize that we do not argue that these processes are critical for the maintenance of adherens junctions. Our results lead us to the conclusion that a strong connection between the actin cytoskeleton and adherens junctions through vinculin is necessary for the proper progression of all these functions. The choice of the 24hpe timing was made because it is a state in which vinculin function was reduced without affecting the stability of adherens junctions (this extent is now further supported by the new results shown in Figure S3), thus allowing us to exclude an effect of the lack of adherens junctions on the mentioned processes.

R

The paper needs to have a schematic diagram to clearly summarize the relationship of all the processes and vinculin at a single cell level.

A

We agree with the reviewer that a summary graph will help understand the results of the work. We have added a graphical abstract as a summary of the work.

R

Below are my comments on individual figures:

Fig 1.

1J, J': In J, there are two very strong F-actin stripes across the middle of the transfected neural tube and it seems there are more cells also. Consistently, J' shows a redistribution of F-actin on the transfected side. I am curious it is a new function of vinculin, or it is a secondary effect from other defective processes.

A

As the reviewer rightly points out, the suppression of vinculin at long times, more than 36 hours, causes the accumulated actin at the apical pole, where it forms apical rings, to occupy more basal positions, while there is a disorganization of the neuroepithelium and many cells become loose within the ventricle. As we have already mentioned in other places in this manuscript, our hypothesis is that vinculin is essential to maintain the firmness of the apical actin rings, and when vinculin is suppressed, initially small defects occur that do not affect the overall structure of actin. However, eventually, the entire actin scaffold collapses, and the apical rings fail to keep the cells from protruding into the ventricle.

R

Fig 2.

2B: It seems there are more vinculin proteins on the apical side, but they are very sparsely distributed. Is that due to the dynamic regulation of the polarized distribution of vinculin or the transfection efficiency? If adherens junction is the anchor for cell attachment and vinculin is an important component of it, the cells highly expressing VCL-mCherry on the apical side should have more active BAINM. This can be tested by live imaging on the cells expressing VCL-mCherry and H2B-GFP.

A

The difference in apical accumulation between the cells shown in Figure 2B is due to differences in transfection efficiency, as evidenced by the ratio between cytoplasmic and apical VCL within the same cells. In other words, cells with low apical VCL do not show cytoplasmic VCL, whereas cells with cytoplasmic VCL have a higher apical presence. The overexpression of VCL does not affect the BAINM, as demonstrated in Figure 8C where we show that expression of VCL does not modify the mean nuclear position of the transfected cells (ShControl + pCIG occupies position 47.55 and ShControl + VCL occupies position 48.69) . It is likely that the reinforcement between the apical actin junctions and AJs is essential for BAINM, but an excessive reinforcement does not have an effect. This supports our model that vinculin or actomyosin do not exert force on their own, but rather act as reinforcements to support the force exerted by the tubulin cytoskeleton.

R

2C: The optical resolution is not sufficient to argue a colocalization between vinculin and Ncad. To clearly show vinculin is the component of adherens junction, it is important to show one high resolution image (using 63x objective lens with high NA) of VCL-mCherry and N-Cad staining at single cell level.

A

We agree with the reviewer that those images needed higher resolution to claim that vinculin was present in the adherens junctions. However, we are confident that the new images included in the revised work, captured using a 100X lens, showing the co-localization of vinculin with β -catenin (Figure 2F) and N-cadherin (Figures 8C and 9A), will fully satisfy the reviewer.

R

Fig 4-6.

In Materials and Methods, it is stated the live imaging was performed 8-10 hours after transfection of shVCL. This should be clearly stated in Results so the readers know how to interpret the data.

A

Thank you, it has been added.

R

Need to add scale bars to the images in 4A-C, 5A-C and all videos.

A

Thank you, it has been added.

R

The authors show velocity and directionality changes in myosin and vinculin knock down cells, but I also observed another interesting phenomenon. In video 5 (BAINM in Bleb treated cells), the authors label one nuclei that was moving around in the middle of the tissue, but below this nuclei there was another one which also initially moved around in the middle of the tissue and then moved smoothly to the apical side from 4:05-4:35. The moving behavior of the second nuclei in this video is similar the shVCL transfected cell in video 6 (BAINM in shVCL treated cells). In contrast, the movement of the normal cells is smoother in video 4. This is consistent with the trajectory analysis. If you compare 5F with 5G and H, indeed the normal cells have straighter trajectories, while the cells treated with Bleb or expressing shVCL show zigzag trajectories and become a little bit straighter over time. Is it possible that this phenotype is more directly relevant to the absence of vinculin while abnormal cell velocity and directionality are the secondary effects?

A

In our opinion, and in agreement with the reviewer, we believe that the changes in speed and directionality are actually consequences of the abnormal trajectories exhibited by cells in which vinculin has been suppressed.

R

Or could it be just due to a compression of neighboring cells?

A

We do not believe so because the cells accumulate on the basal side, which means that compression is greater the further away the cells are from the apical side.

R

Why do the nuclei move more straighterly later on?

A

Possibly because over time, tubulin fibers shorten more and more, increasing tension to the point where it becomes sufficient to propel the nucleus. It's like the actin acts as an elastic band, and the tubulin acts as a ratchet, progressively increasing tension.

R

Are the zigzag trajectories caused by the defective internalization of centrosomes to pull the nuclei through tubulin or does it due to the loose adherens junction?

A

The zigzag pattern is due to the lack of tension in the actin ring caused by the absence of vinculin. Even though the tubulin shortens, the lack of tension in the actin prevents the nucleus from advancing. It's like an old elastic band that no longer has tension. The failure

of centrosome internalization is another consequence of the lack of tension in the apical actin ring.

R

Is the interaction between vinculin and myosin direct or indirect?

A

To the best of our knowledge, vinculin does not bind to myosin, but rather it binds to actin. However, it is necessary for actin to be anchored at adherens junctions with vinculin reinforcement in order for myosin to carry out its contractile function.

R

Whether and how this migration behavior affects adherens junctions?

A

Addressing the reviewer's concern regarding whether aberrations in nuclear movement caused by vinculin suppression could be affecting adherens junctions, we conducted a series of experiments by transfecting ShVCL for 24 hours (that includes the entire period of nuclear movement recording) and capturing open book images of N-cadherin and phalloidin staining. We were able to verify that, at least up to that point, the structure of adherens junctions remained fully preserved in the transfected cells. These results are mentioned in the Results section and constitute the new Figure S3.

R

In my view, addressing any of these questions using the imaging system the authors established will deepen our understandings of the cell biology in vivo.

Fig 7.

I am a little bit disappointed that the authors did not use vinculin as a key reagent to further test the relationships between cell cycle progression, centrosome internalization, and INM.

A

At this point, we cannot agree with the reviewer. We did utilize suppression and overexpression of vinculin to study its effect on the INM and cell cycle, and we verified that overexpression does not affect either the INM (Figure 9K) or the cell cycle (Figures 9L and M-S).

R

7L-K: I have another hypothesis: one centrosome internalization may establish tubulin (one end binds to the centrosome connecting to the nuclei; the other end binds to the centrosome connecting to the adherens junction) to pull the nucleus to the apical side. Is it possible that in the vinculin knock down cells the failure of centrosome internalization blocks the formation of this tubulin structure; as a result, the nuclei show a defective and zigzag BAINM?

Certainly! Here's an improved translation:

A

The hypothesis put forward by the reviewer is indeed intriguing. However, it is not necessary for the centriole to generate a new tubulin cytoskeleton. In fact, various studies referenced in the manuscript demonstrate the presence of a tubulin cytoskeleton throughout the entire cell during interkinetic nuclear migration (INM). Furthermore, these studies highlight the role of the tubulin cytoskeleton in exerting nuclear pulling forces. In fact, we initially depicted this distribution in the graphical abstract, which regrettably was

not included in the previous submission. We sincerely apologize for any confusion caused, as we understand this may have hindered the comprehension of the mechanism we propose.

To facilitate a clearer understanding of our proposed mechanism, we have now included the graphical abstract. Specifically, our findings indicate that both the basal-apical aspect of INM and the internalization of the centriole require vinculin to reinforce the apical actin cytoskeleton. This reinforced cytoskeleton serves as an anchoring point for microtubules, which exert the necessary forces to facilitate nuclear traction.

Once again, we apologize for any confusion caused and hope that the inclusion of the graphical abstract will enhance the comprehension of our proposed mechanism.

R

Fig 9.

Does disrupting the normal structure of adherens junctions in Ncad mutants produces similar phenotype?

A

Possibly, as the actin cytoskeleton requires the presence of adherens junctions to support itself. Unfortunately, this experiment cannot be conducted since, as demonstrated in our recent publication in JCB Herrera et al., 2021, the suppression of N-cadherin leads to a rapid collapse of cellular polarity and, consequently, the neuroepithelial structure, making it impossible to study the nuclear movements characteristic of interkinetic nuclear migration (INM).

R

The experiments in Fig3-7 were performed in the embryos where the adherens junctions still have normal structure (apical side of the transfected neural tube looks normal). The phenotypes in INM and cell cycle progression observed in those experiments suggest some other functions of vinculin rather than its binding to adherens junctions, are required for these cellular processes. Or is it possible that in those experiments the adherens junction structure looks normal but is loose? If so, the authors need to show that.

A

The experiments with vinculin A50I exclude the involvement of other functions of vinculin, apart from its binding to adhesion proteins, in these processes. We believe that this is due to a loose and low-tension apical actin ring and adherens junctions (AJs). Particularly, because shortly afterward, the tension decreases even further, and the apical actin cytoskeleton becomes unable to keep the cells within the epithelium. We agree with the reviewer that studying the sequence of tension loss in the apical actin ring would be very interesting. However, in our opinion, it is beyond the possibilities of the present work.

September 26, 2023

RE: JCB Manuscript #202106169R-A

Dr. Sebastián Pons
Molecular Biology Institute of Barcelona
Department of Cell Biology
C/ Baldiri Reixac, 10-12
Edificio Cluster, Parc Científic de Barcelona
Barcelona, Barcelona 08028
Spain

Dear Dr. Pons,

Thank you for submitting your revised manuscript entitled "Vinculin is required for interkinetic nuclear migration (INM) and cell cycle progression." We would be happy to publish your paper in JCB pending final revisions necessary to meet our formatting guidelines and minor textual changes to address final reviewer comments.

The revised manuscript was re-assessed by two of the original reviewers who agree that the work has been greatly improved. The third reviewer was not available to re-review. Reviewer #1 does raise an important point about the absence of direct visualization of endogenous vinculin. Based on additional data that you provided to us we understand that you have tried but have not been able to find an antibody to vinculin that works on chicken embryos. Thus additional data will not be required but we do ask that you address the reviewer comments with text revisions and discuss limitations and caveats of your results.

A. MANUSCRIPT ORGANIZATION AND FORMATTING:

1) Text limits: Character count for Articles is < 40,000, not including spaces. Count includes title page, abstract, introduction, results, discussion, and acknowledgments. Count does not include materials and methods, figure legends, references, tables, or supplemental legends.

2) Figure formatting: Articles may have up to 10 main text figures. Scale bars must be present on all microscopy images, including inset magnifications. Please move the scale bar in Figure 2A to the bottom of the image. Molecular weight or nucleic acid size markers must be included on all gel electrophoresis. The labels should correspond to the molecular weight standards that were run on the particular gel and not the expected size of the protein or nucleic acid of interest.

Also, please avoid pairing red and green for images and graphs to ensure legibility for color-blind readers. If red and green are paired for images, please ensure that the particular red and green hues used in micrographs are distinctive with any of the colorblind types. If not, please modify colors accordingly or provide separate images of the individual channels.

3) Statistical analysis: Error bars on graphic representations of numerical data must be clearly described in the figure legend. The number of independent data points (n) represented in a graph must be indicated in the legend. Please, indicate whether 'n' refers to technical or biological replicates (i.e. number of analyzed cells, samples or animals, number of independent experiments). If independent experiments with multiple biological replicates have been performed, we recommend using distribution-reproducibility SuperPlots (please see Lord et al., JCB 2020) to better display the distribution of the entire dataset, and report statistics (such as means, error bars, and P values) that address the reproducibility of the findings.

Statistical methods should be explained in full in the materials and methods. For figures presenting pooled data the statistical measure should be defined in the figure legends. Please also be sure to indicate the statistical tests used in each of your experiments (both in the figure legend itself and in a separate methods section) as well as the parameters of the test (for example, if you ran a t-test, please indicate if it was one- or two-sided, etc.). Also, if you used parametric tests, please indicate if the data distribution was tested for normality (and if so, how). If not, you must state something to the effect that "Data distribution was assumed to be normal but this was not formally tested."

4) Materials and methods: Should be comprehensive and not simply reference a previous publication for details on how an experiment was performed. Please provide full descriptions (at least in brief) in the text for readers who may not have access to referenced manuscripts. The text should not refer to methods "...as previously described."

5) For all cell lines, vectors, constructs/cDNAs, etc. - all genetic material: please include database / vendor ID (e.g., Addgene, ATCC, etc.) or if unavailable, please briefly describe their basic genetic features, even if described in other published work. If gifted to you by other investigators then provide references where appropriate. Please be sure to provide the sequences for all of your oligos: primers, si/shRNA, RNAi, gRNAs, etc. in the materials and methods. You must also indicate in the methods the source, species, and catalog numbers/vendor identifiers (where appropriate) for all of your antibodies, including secondary. If antibodies are not commercial, please add a reference citation if possible.

6) Microscope image acquisition: The following information must be provided about the acquisition and processing of images:

- a. Make and model of microscope
- b. Type, magnification, and numerical aperture of the objective lenses
- c. Temperature
- d. Imaging medium
- e. Fluorochromes
- f. Camera make and model
- g. Acquisition software
- h. Any software used for image processing subsequent to data acquisition. Please include details and types of operations involved (e.g., type of deconvolution, 3D reconstitutions, surface or volume rendering, gamma adjustments, etc.).

7) References: There is no limit to the number of references cited in a manuscript. References should be cited parenthetically in the text by author and year of publication. Abbreviate the names of journals according to PubMed.

8) Supplemental materials: There are strict limits on the allowable amount of supplemental data. Articles may have up to 5 supplemental figures and 10 videos. Please also note that tables, like figures, should be provided as individual, editable files. A summary of all supplemental material should appear at the end of the Materials and methods section. Please include one brief sentence per item.

9) Videos: For publication the journal requires MP4 files no larger than 20 MB. For optimal compatibility across operating systems and devices, please select H.264 compression when saving. Video legends should describe what is being shown, the cell type or tissue being viewed (including relevant cell treatments, concentration and duration, or transfection), the imaging method (e.g., time-lapse epifluorescence microscopy), what each color represents, how often frames were collected, the frames/second display rate, and the number of any figure that has related video stills or images.

10) eTOC summary: A ~40-50 word summary that describes the context and significance of the findings for a general readership should be included on the title page. The statement should be written in the present tense and refer to the work in the third person. It should begin with "First author name(s) et al..." to match our preferred style.

11) Conflict of interest statement: JCB requires inclusion of a statement in the acknowledgements regarding competing financial interests. If no competing financial interests exist, please include the following statement: "The authors declare no competing financial interests." If competing interests are declared, please follow your statement of these competing interests with the following statement: "The authors declare no further competing financial interests."

12) A separate author contribution section is required following the Acknowledgments in all research manuscripts. All authors should be mentioned and designated by their first and middle initials and full surnames. We encourage use of the CRediT nomenclature (<https://casrai.org/credit/>).

13) ORCID IDs: ORCID IDs are unique identifiers allowing researchers to create a record of their various scholarly contributions in a single place. Please note that ORCID IDs are required for all authors. At resubmission of your final files, please be sure to provide your ORCID ID and those of all co-authors.

14) Please note that JCB now requires authors to submit Source Data used to generate figures containing gels and Western blots with all revised manuscripts. This Source Data consists of fully uncropped and unprocessed images for each gel/blot displayed in the main and supplemental figures. Since your paper includes cropped gel and/or blot images, please be sure to provide one Source Data file for each figure that contains gels and/or blots along with your revised manuscript files. File names for Source Data figures should be alphanumeric without any spaces or special characters (i.e., SourceDataF#, where F# refers to the associated main figure number or SourceDataFS# for those associated with Supplementary figures). The lanes of the gels/blots should be labeled as they are in the associated figure, the place where cropping was applied should be marked (with a box), and molecular weight/size standards should be labeled wherever possible. Source Data files will be directly linked to specific figures in the published article.

15) Journal of Cell Biology now requires a data availability statement for all research article submissions. These statements will

be published in the article directly above the Acknowledgments. The statement should address all data underlying the research presented in the manuscript. Please visit the JCB instructions for authors for guidelines and examples of statements at (<https://rupress.org/jcb/pages/editorial-policies#data-availability-statement>).

B. FINAL FILES:

Thank you for your attention to these final processing requirements. Please contact the journal office with any questions, cellbio@rockefeller.edu.

Thank you for this interesting contribution, we look forward to publishing your paper in Journal of Cell Biology.

Sincerely,

Marianne Bronner, PhD
Monitoring Editor
Journal of Cell Biology

Dan Simon, PhD
Scientific Editor
Journal of Cell Biology

Reviewer #1 (Comments to the Authors (Required)):

The authors have addressed adequately most of my previous concerns. The inclusion of higher magnification images in this new version of the manuscript makes the conclusions more compelling. However, I am still not convinced about the justification for the use of the Vinculin T12 mutant to analyse the normal localization of Vinculin. It has been shown that this mutant affects Vinculin dynamic and leads to exaggerated focal adhesions (Cohen et al., 2015); if so, how is it possible to conclude about the normal endogenous localization of Vinculin using this mutant? The authors need to clarify this important issue, as most of their conclusions are based on Vinculin localization.

In addition, the authors need to include in the manuscript a statements about the limitations of the work as clear as the one included in the rebuttal letter: "we must admit that with the tools and data presented, a direct effect on the focal adhesions of the basal pole cannot be definitively ruled out, which could indirectly interfere with interkinetic nuclear migration or centriole internalization."

Reviewer #2 (Comments to the Authors (Required)):

The authors have made appropriate integrations of some of the missing controls and data validations and have significantly improved the overall structure of the manuscript. Thus, the quality of the paper has been raised to a sufficient level to warrant publication in JCB.